# A Method for Constructing Directional Surface Wave Spectra from ICESat-2 Altimetry

Momme C. Hell[1] and Christopher Horvat[1,2]

[1]Brown University, Providence, RI, USA

[2]University of Auckland, Auckland, NZ

**Correspondence:** Momme Hell (mhell@brown.edu)

**Abstract.** Sea ice is important for Earth's energy budget as it influences surface albedo and air-sea fluxes in polar regions. On its margins, waves heavily impact sea ice. Routine and repeat observations of waves in sea ice are currently lacking, and therefore a comprehensive understanding of how waves interact with sea ice and are attenuated by it is elusive.

In this paper, we develop methods to separate the two-dimensional (2D) surface wave spectra from sea ice height observa-
tions made by the ICESat-2 laser altimeter, a polar-orbiting satellite. A combination of a linear inverse method, called Gener-
alized Fourier Transform (GFT), to estimate the wave spectra along each beam and a Metropolitan Hasting (MH) algorithm
to estimate the dominant wave's incident angle was developed. It allows us to estimate the 2D wave signal and its uncertainty
from the high-density, unstructured ATL03 ICESat-2 photon retrievals. The GFT is applied to re-binned photon retrials on 25
km segments for all six beams and outperforms a discrete Fourier transform in accuracy while having fewer constraints on the
data structure.

The MH algorithm infers wave direction from beam pairs every 25 km using coherent crests of the most energetic waves.
Assuming a dominant incident angle, both methods together allow a decomposition into 2D surface wave spectra with the
advantage that the residual surface heights can potentially be attributed to other sea ice properties. The combined GFT-MH
method shows promise in routinely isolating waves propagating through sea ice in ICESat-2 data. We demonstrate its ability on
a set of example ICESat-2 tracks, suggesting a detailed comparison against in-situ data is necessary to understand the quality
of retrieved spectra.

## 1   Introduction and Problem Description

Sea ice covers up to 9% of the world's oceans, and plays an important role in the energy balance of Earth's climate. Even
though sea ice damps ocean surface waves (Squire, 2007), broad regions along the periphery of the sea-ice-covered ocean are
continually under the influence of surface waves (Rapley, 1984; Horvat et al., 2020; Thomson, 2022; Horvat, 2022). These

regions are collectively referred to as the Marginal Ice Zone (MIZ). In the MIZ, waves influence sea ice's thermodynamic and dynamic properties and impact the coupled exchange between atmosphere and ocean. Currently, we do not have reliable global observations of waves in sea ice, and hence are unable to sufficiently understand air-sea exchange and wave propagation in the MIZ. This paper describes how ICESat-2 altimeter observations can be used to record wave spectra in the MIZ, and to infer additional sea ice properties for building parametrizations of wave attenuation in sea ice.

Models of wave propagation in sea ice typically evolve the ocean surface wave spectrum, $\tilde{S}_h(k)$ (meter$^2$ $k^{-1}$, $k$ is the wavenumber), which is attenuated when it comes into contact with sea ice. There has been significant debate over the functional form and dependencies of this attenuation (Squire, 2018; Thomson et al., 2021). Yet as it controls how deep waves reach into sea ice, it is vital for modeling MIZ variability and coupled feedbacks in the polar seas.

Constraining ice-induced wave attenuation is challenging because wave observations in ice are difficult to make at scale. A majority of observations of waves in ice are carried out using ships or arrays of floating buoys deployed by ships (or by helicopters from ships, see, for example Thomson, 2022, and references therein). While such observations provide high temporal frequency observations of wave spectra, they only cover a limited geographic domain, and are limited by the sea ice types and conditions at the original buoy locations. Recently satellite remote sensing technologies have shown promise for describing wave spectra in sea ice regions. SAR imagery is capable of observing wave crests as they move into the MIZ, and the two-dimensional wave spectrum can be constructed in good agreement with in-situ-observed spectra if the sea ice is not rough (Stopa et al., 2018; Ardhuin et al., 2017). However, SAR alone cannot observe continuous spectra as they propagate into the sea ice.

The ICESat-2 (IS2) altimeter has the potential to greatly increase the quantity of available observations of wave-ice interactions, either alone or in combination with other remote sensing instruments (Collard et al., 2022). ICESat-2 carries a single measurement tool, ATLAS, a six-beam laser oriented in three weak/strong pairs (Fig. 1a, colored lines) offset at a near-uniform three kilometers on the ground, with a weak-strong beam lateral offset of about 90m meters. ATLAS measures the return time of individual photons to infer the height of the ice/ocean surface. Typical along-track photon spacings can be centimeters or smaller, and so IS2 is capable of directly sampling ocean surface waves, particularly over reflective sea ice.

Recent studies have examined waves in sea ice using IS2, basing their results on a higher-order sea ice height product derived from photon retrievals (known as ATL07). Horvat et al. (2020) identified the capability of IS2 to retrieve ocean waves by examining a storm in the Barents Sea in 2019 and used a simple threshold to establish where and when waves were observed in the sea ice to produce global maps of the MIZ. In Collard et al. (2022), IS2 retrievals during this Barents Sea storm were shown to compare well with model and SAR-based observation data. Brouwer et al. (2021) selected a series of Southern Hemisphere IS2 retrievals, analyzing wave attenuation using direct spectral transform methods. Both found that areas affected by waves were common in both hemispheres, with repeated measurements of waves hundreds of kilometers into the sea ice zone, particularly in the Southern Hemisphere.

Three challenges limit the direct comparison of IS2-derived wave spectra to observations and models. First, waves propagate at an angle $\theta$ relative to the along-track direction of the satellite (Fig. 1b), and observable wave lengths $\lambda$ are aliased by an unknown factor $\cos{(\theta)}^{-1}$ (Rapley, 1984; Horvat et al., 2019; Yu et al., 2021). Second, observed surface height variability is a

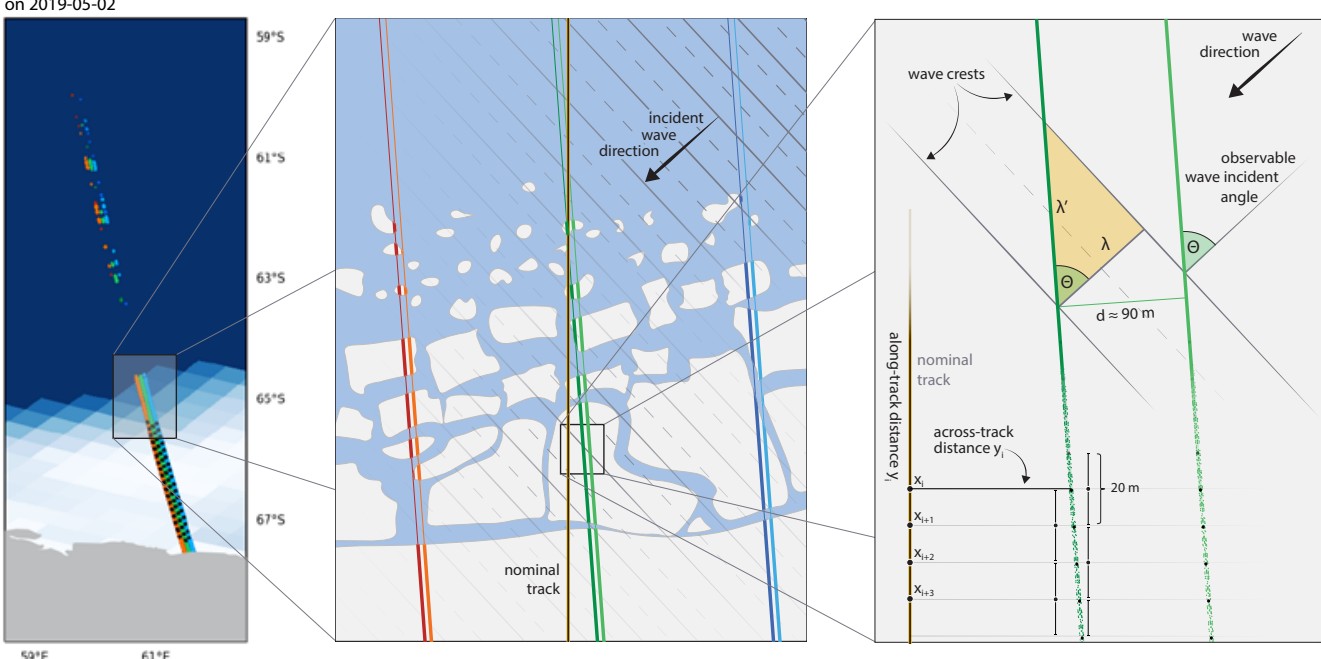

**Figure 1.** Illustration of ICESat-2 (IS2) beams intersecting the Marginal Ice Zone (MIZ) under the presence of waves. a) ICESat-2 data and the CDR Sea Ice Concentration for track 05160312 on May 2nd 2019. The three beam pairs are shown as red/orange (gt1l/gt1r), dark/light green (gt2l/gt2r), or dark/light blue (gt3l/gt3r) lines. The black dots show the segment positions 12.5 km apart (define in section 2.3). b) Schematic of the IS2 beams that observe sea ice and surface waves in the MIZ. The vertical black line is the nominal IS2 reference ground track. The incident waves come from the top-right along the black arrow with wave crests (solid) and valleys (dashed). c) Details view of an ideal monochromatic wave observation by IS2 in sea ice. The IS2 beam pair (light/dark green lines) observes an incident wave (along the black arrow) of wavelength $\lambda$ with an angle $\theta$ as $\lambda'$. With the beam-pair distance $d$, one can calculate $\theta$ from the phase lag of the incident wave crests (sec. 3.1). The along-/across-track coordinate system is referenced to the nominal track, while each data point is the weighted mean of a 20-meter stencil (sec. 2.1).

convolution of the dynamic ocean topography, sea ice topography, surface waves, and noise. The surface wave signal can then be successfully reconstructed if these other signals are on a different scale, like dynamic ocean topography, or not periodic, like the sea ice topography and noise. Third, the fractured nature of sea ice, the influence of clouds, and changing surface albedo cause gaps or irregularities in IS2 photon retrieval rates, creating a high-density but irregular observation. The method must
be applicable to irregular data without generating spurious sources of variance, i.e., artificial wave energy. The above factors complicate direct assessments of spectra and their attenuation in sea ice.

Here we demonstrate a method for producing angle-corrected, two-dimensional (2D) wave spectra in sea ice using photon height data from ICESat-2. We partition surface height variability into waves and sea-ice or noise-related components. It permits a direct assessment of the most significant wave energy along each track to record wave attenuation and evaluate
numerical attenuation schemes. We show this partitioning allows for significantly improved sea-ice height estimates in the MIZ, and may also allow for expanding existing higher-level IS2 products to broader ice-covered regions.

In this paper, we demonstrate this method on four example cases, Track 1 to 4 (details in suppl. table. Their granule, i.e. their identification number, is also given in each figure). We describe the pre-processing of IS2 along-track photon heights in section 2.1 and develop a harmonic fitting procedure applied to individual IS2 beam in section 2.2 (the GFT method).
Subsequently, we develop a multi-beam, Monte-Carlo method for bias-correcting along-track wavelengths in section 3 (the MH method), which enables us to provide two-dimensional wave spectra derived from along-track data in section 3.2, and a decomposition of photon variance in section 4. In section 5, we discuss the limitations and assumptions of the proposed methods and conclude in section 6 how they can be used to develop improved models of attenuation of waves in sea ice.

## 2 Along-track Wave Spectra from IS2

The primary aim of this analysis is to assess surface height variability in the MIZ. Hence we want to use the highest data resolution we can handle, though we are agnostic about the classification of photon returns. That is the L2-level product ATL03 from Neumann et al. (2021). For comparison, we show the photon cloud data from ATL03 and the surface heights and type classifications from the higher level ATL07/10 product in figure 2 as dark blue, light blue, or orange dots (Kwok et al., 2021). By requiring 150 consecutive photons to identify a sea ice segment, the ATL07 product accounts for most of the height
variability from the ATL03 product. Yet it misses retrievals in the MIZ (suppl. Fig. 2, white and gray area) and within the sea ice (Fig. 2). For better resolution, the following analysis is based on the photon cloud data from ATL03.

### 2.1 Data Pre-Processing

Linearly inverting photon data requires exact along and across-track information about photon positions. Along-track photon positions are first re-referenced to the most equatorward position on the nominal ATLAS ground track (Fig. 1b, black line). The
85 most equatorward position is evaluated from the ATL07 (Kwok et al., 2021) product and set to the beginning of the 1st 100 km of along-track data where there is an average of at least 0.02 photons per meter (defined as $X = 0$ throughout the paper). This threshold and re-referencing are used to exclude large areas of nearly no data in the transition zone between the open ocean and MIZ. All tracks are then followed in a poleward direction, until the variance of any of the 6 beams exceeds a factor of 10 times the variance of the first 15% on the equatorward side of the track (suppl. Fig. 2, dashed black lines). This avoids including
90 impacts from coastal or land ice around the Antarctic coast. The redefined along-track direction $x$ and an across-track direction $y$ are then used as the coordinate system throughout the analysis (Fig. 1b, suppl. Fig. 1 and 2).

After removing the cumulative surface height correction (`dem_h` taken from the ATL07/10 dataset, Kwok et al., 2021), we bin photon measurements into 20-meter stencils that overlap by 50%. This yields a 10-meter along-track resolution (Fig. 2 green line). Note this differs from the procedure used to form sea ice surface heights in the ATL07 product, which averages
height data for each set of 150 photons along-track. ATL07 has a constant photon count, with the trade-off of irregularly spaced segments of varying length in the MIZ, while our approach provides more regularly-spaced data with the trade-off of having a variable photon count in each stencil and potentially including retrievals of water near sea ice. Stencils with fewer than five photons are excluded, which also leads to data gaps corresponding to no or very low photon retrievals due to sea-ice leads,

open water, clouds, or other noise (suppl. Fig. S1). These data gaps also lead to an irregularity in the data, but here each stencil
mean represents the same area and hence better captures the wave phase.

A mean photon height $h_c(x)$ is calculated in each 20-meter stencil as the mean of the photons weighted by their inverse distance to the stencil center, using a Gaussian weighting function with a standard deviation of 10m. We tested other data reduction methods, like using the median or mode of the stencil, finding results insensitive to the choice of the binning method (suppl. Fig. S2). The same 20-meter stencil also provides an uncertainty estimate $\sigma_h(x)$ (proportional to Fig. 2 blue area)
representing the varying photon density. This uncertainty is used to define the data prior (sec. A1) for the harmonic inversion in section 2.2.

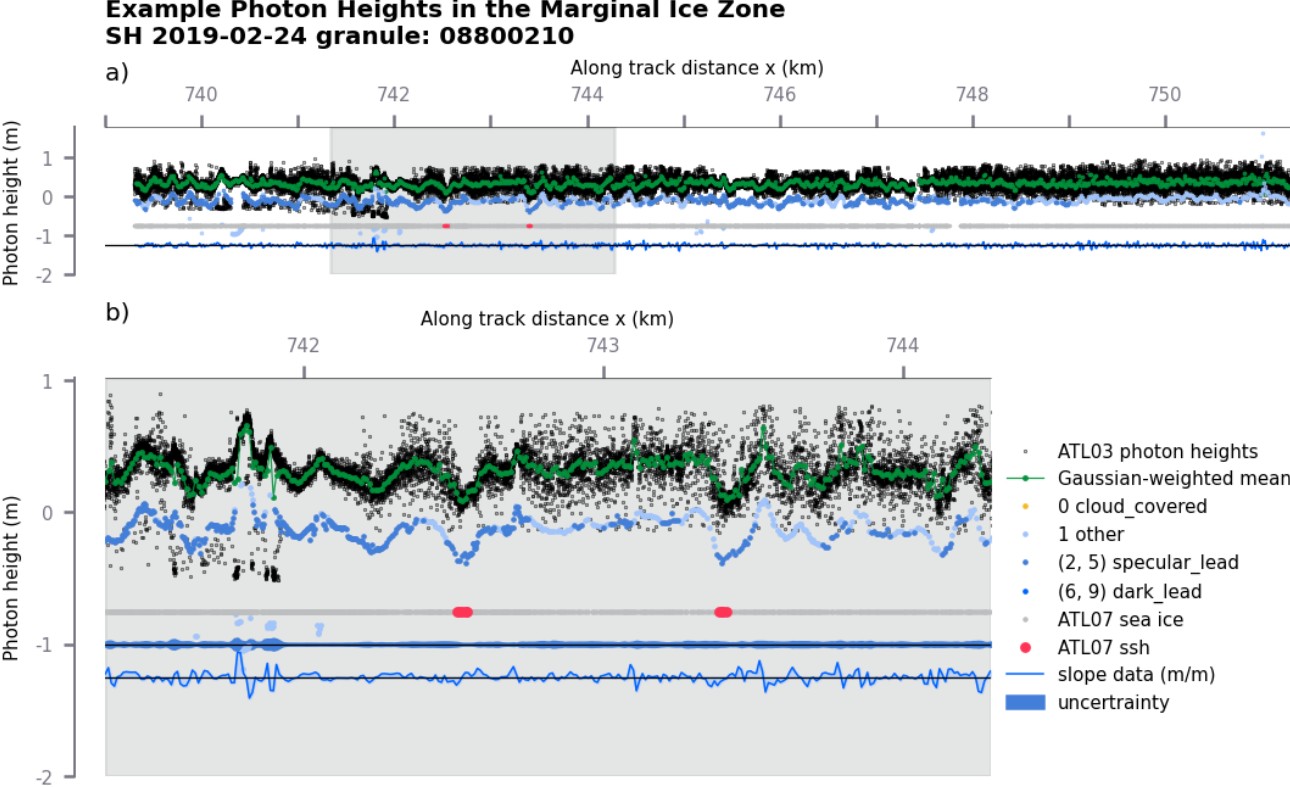

**Figure 2.** ATL03 photon cloud and ATL07 surface heights for example track 1. Individual photons are shown in black, and the 20-meter weighted average as a green line. The ATL07 photon heights are shown as light blue, dark blue, or black dots, with the color corresponding to their type category provided in ATL07. The ATL07 sea ice and sea ice lead classification is illustrated along y = -0.75 as gray and red segments. The surface slope based on the weighted photon average is shown as a blue line with a -1.25 offset, and its uncertainty estimate is shown as blue shading with an offset of $y = -1$. The panel b) is an inset of the gray area in panel a).

The re-sampled mean photon height data is used to calculate a series of along-track surface slopes (Fig. 2 thin blue line) by taking the along-track derivative and applying a spike-removing algorithm. Using the along-track surface slope data focuses the timeseries analysis on local photon-height changes rather than the magnitude of the total surface height field. The spike-

removal reduces peaks in the slope data, which can be from sea-ice height changes or especially those resulting from ice-ocean transitions. Because the slope field approximates the derivative of the height field, the spectrum of slopes $\hat{S}_c(k)$ is readily connected to the spectrum of surface heights $\tilde{S}_h(k)$, as $\tilde{S}_h(k) = k\tilde{S}_c(k)$. The surface height field can therefore be directly reconstructed from the slope spectrum, as we show below. The generalized Fourier transform (sec. 2.2) and directional estimates (sec. 3.1) are then applied on 25-km long segments of these surface slopes, with uncertainty estimates (Fig. 2). The 25-km segments also overlap by 50%, providing an along-track spectral estimate every 12.5 km.

## 2.2 Generalized Fourier Transform (GFT)

We estimate along-track wave spectra using a Generalized Fourier Transform (GFT). The GFT is a harmonic fit of sin- and cos-pair coefficients, which together determine amplitude and phase at each wavenumber. The model complexity is defined by the number of resolved wave numbers and its success depends on prior (assumed) knowledge about the data's uncertainty and model structure.

We use a GFT to overcome several disadvantages appearing when implementing a standard Discrete Fourier Transform (DFT) to unstructured data. While the DFT is a fast variance-conserving algorithm, it requires periodic, continuous, and equally-spaced data. The DFT implies that frequency bands are harmonics over a domain or segment length $L$, an arbitrary limitation on the resolved frequencies. To make segments periodic, often tapering or windowing is applied to the segmented data. In addition to the data's non-periodicity, the common presence of data gaps in IS2 retrievals requires extrapolation or gap-filling to create continuous, equally spaced data suitable for a DFT. These both lead to commonly known problems of the DFT, like energy leaks/compensation in spectral space. The data handling needed for DFT can erode the signal substantially, especially in the MIZ (Fig. 3 b,c gray and green lines).

The GFT method outlined below works on any grid, incorporates data uncertainty, and does not require periodicity. The GFT can be customized to the frequencies of interest with the additional benefit of providing a standard error in real and wavenumber space.

## 2.3 Harmonic wave inversion

We follow Wunsch (1996), Menke (2018), and Kachelein et al. (2022), using generalized least squares to derive spectral amplitudes in each 25-km segment $X_i$ (Fig. 4). Slope data in each section $X_i$ is a series of unevenly spaced mean-zero data points and is expressed as a column vector $\mathbf{b} = \partial_x \mathbf{z}$ of length $N_i$ where $\mathbf{z}$ are the height data. These data are then modeled as the sum of sinusoids with wave numbers in the range of swell and wind waves

$$\mathbf{b} = \mathbf{H}\,\mathbf{p} + \mathbf{r}, \tag{1}$$

where $\mathbf{H}$ is an $N_i \times 2M$ regressor matrix of basis functions, $\mathbf{p}$ is the model parameter vector of length $2M$, and $\mathbf{r}$ is the vector of the residual timeseries of length $N_i$. The columns of $\mathbf{H}$ are sines and cosines of prescribed wave numbers, $k'_m = 2\pi/\lambda_m$ for wavelengths $\lambda_m$ indexed by $m = 1, 2, ..M$. We use prime notation here and throughout the paper to indicate observed wave

variables along the direction of each beam and unprimed notation for variables in the direction of the traveling wave (see Fig. 1c). The problem can then be written as

$$\mathbf{b} = \sum_{m=1}^{M} \left( a_m \cos\left(k'_m \mathbf{x}\right) + c_m \sin\left(k'_m \mathbf{x}\right) \right) + \mathbf{r}, \tag{2}$$

$$\tag{3}$$

with model parameters,

$$\mathbf{p} = [a_1, a_2, ...a_M, c_1, ...c_M]^T, \tag{4}$$

at positions

$$\mathbf{x} = [x_1, x_2, ..., x_{N_i}]^T. \tag{5}$$

To find the most probable $\mathbf{b}$ given a set of model parameters $\mathbf{p}$, we need to estimate the autocovariance matrices of the residual

$\mathbf{R} = \left\langle \mathbf{r}\mathbf{r}^T \right\rangle$, i.e. the error of the data, and the autocovariance matrix of the model $\mathbf{P} = \left\langle \mathbf{p}\mathbf{p}^T \right\rangle$, where $(\langle \cdot \rangle)$ is the expected value. Then the most probable estimate of the data $\mathbf{b}$ can be found by estimating the maximum of the posterior probability distribution $P(\mathbf{p}|\mathbf{b})$. Using Bayes' theorem (Kachelein et al., 2022), or, alternatively the matrix inversion lemma (Wunsch, 1996), given the data $|\mathbf{b}$, the most likely estimate of the model parameters $\hat{\mathbf{p}}$ is found as,

$$\hat{\mathbf{p}} = \left( \mathbf{H}^T \mathbf{R}^{-1} \mathbf{H} + \mathbf{P}^{-1} \right)^{-1} \mathbf{H}^T \mathbf{R}^{-1} \mathbf{b}. \tag{6}$$

Once the model parameters $\hat{\mathbf{p}}$ are estimated, the model can be expressed in real space using

$$\hat{\mathbf{b}} = \mathbf{H}\,\hat{\mathbf{p}}, \tag{7}$$

(Fig. 3a green lines), or as a power spectrum $\hat{\tilde{\mathbf{S}}}_{GFT}$ (Fig. 3b,c green lines, see suppl. material S1 for derivation). Note that $\hat{\tilde{\mathbf{S}}}_{GFT}$ is substantially different from that of the DFT of the same data (Fig. 3b,c gray and green lines respectively). Gaps in the data, as well as the DFT's requirement for the data to be periodic, creates artificial power in the swell's wavenumber range that

leads to misleading results.

To estimate the model error, the posterior autocovariance of the difference between estimated and true model parameters is defined as the inverse Hessian,

$$\mathbf{Hess}^{-1} = \left\langle (\mathbf{p} - \hat{\mathbf{p}})(\mathbf{p} - \hat{\mathbf{p}})^T \right\rangle = \left( \mathbf{H}^T \mathbf{R}^{-1} \mathbf{H} + \mathbf{P}^{-1} \right)^{-1}, \tag{8}$$

In practice, this is calculated from the "kernel" matrix $\mathbf{H}$, and the (assumed) Gaussian distributed data and model priors $\mathbf{R}$ and $\mathbf{P}$. The trace $\text{tr}(\cdot)$ of the inverse Hessian is then used to estimate the error of the model parameters

$$\hat{\mathbf{p}}_{err} = \text{tr}(\mathbf{Hess}^{-1}), \tag{9}$$

(Fig. 3d, shown in the same units as the power spectra). The error of the fit to the modeled data is also related to the inverse Hessian,

$$\hat{\mathbf{b}}_{err} = (\mathbf{H}^2 \, \mathbf{Hess}^{-1})\mathbf{j}, \tag{10}$$

where $\mathbf{j}$ is a unit vector of length $2M$.

The harmonic inversion of this segment of Track 2 shows how wave spectra can be calculated from strong (gt2r) and weak (gt2l) beams, even if the data has gaps (Fig. 3a). Both beams' spectra are similar in most parts and show a maximum at $k = 0.03$ (Fig. 3b,c). However, the strong beam (gt2r) does show a second local maximum at about $k = 0.06$. Note that the DFT of the same track, and with tapered data, results in a different PDF than the harmonic inversion because of the data gaps.

### 2.3.1 Choice of model resolution and degrees of freedom

The quality of the GFT model depends on the degrees of freedom, the model prior $\mathbf{p}$, and data priors $\mathbf{r}$ (eq. 8). While the number of model parameters, $2M$, remains fixed throughout the analysis, the number of data points in segment $i$, $N_i$, is variable, and may be controlled by the segment length $L$, which is in our case 25 km, with a 12.5km overlap (Fig. 4).

The number of degrees of freedom is $2M - N_i$ and depends on the number of data points in each 25-km segment. A segment with no data gaps and a 10-meter resolution (sec. 2.1) contains $N_i = 2500$ data points. With $2M = 1740$ model parameters, this is an over-determined problem. However, frequent data gaps reduce the data points per segment, which may result in an underdetermined problem with more model parameters than data ($2M > N_i$). The result is then a larger residual $\mathbf{r}$ and larger uncertainty estimate $\hat{\mathbf{p}}_{err}$ (eq. 9 and eq. 10). Even in cases where eq. (1) is underdetermined, we are confident in our wave spectrum estimation within a given error because $\mathbf{P}$ contains prior knowledge about the shape of the solution, i.e., the shape of typical surface wave spectra (sec. 2.4). Most of the segments of the four example tracks in this study have close to 2500 data points and are over-determined (suppl. Fig. S3). Only track 1 and 2 (Fig. 3) are under-determined close to the edge of the ice cover (supply. Fig. S3 a,b).

The choice of segment length also determines the smallest resolvable wavenumber. For example, a segment length of 25 km resolves a wave with a 20s period at an incident angle of $\pm 75°$ about ten times. We set the lowest to-be resolved observed wavenumber to $k_1' = 2.5 \times 10^{-3}$ rad m$^{-1}$ which corresponds to a maximum observed wavelength of 2500 meters (sec. 3.1). The highest wavenumber is chosen as $k_m' = 0.11$ rad m$^{-1}$, a typical period of wind waves of about six seconds. Using evenly spaced wavenumbers with $dk = 1.25 \times 10^{-4}$ results in $M = 869$ wavenumbers.

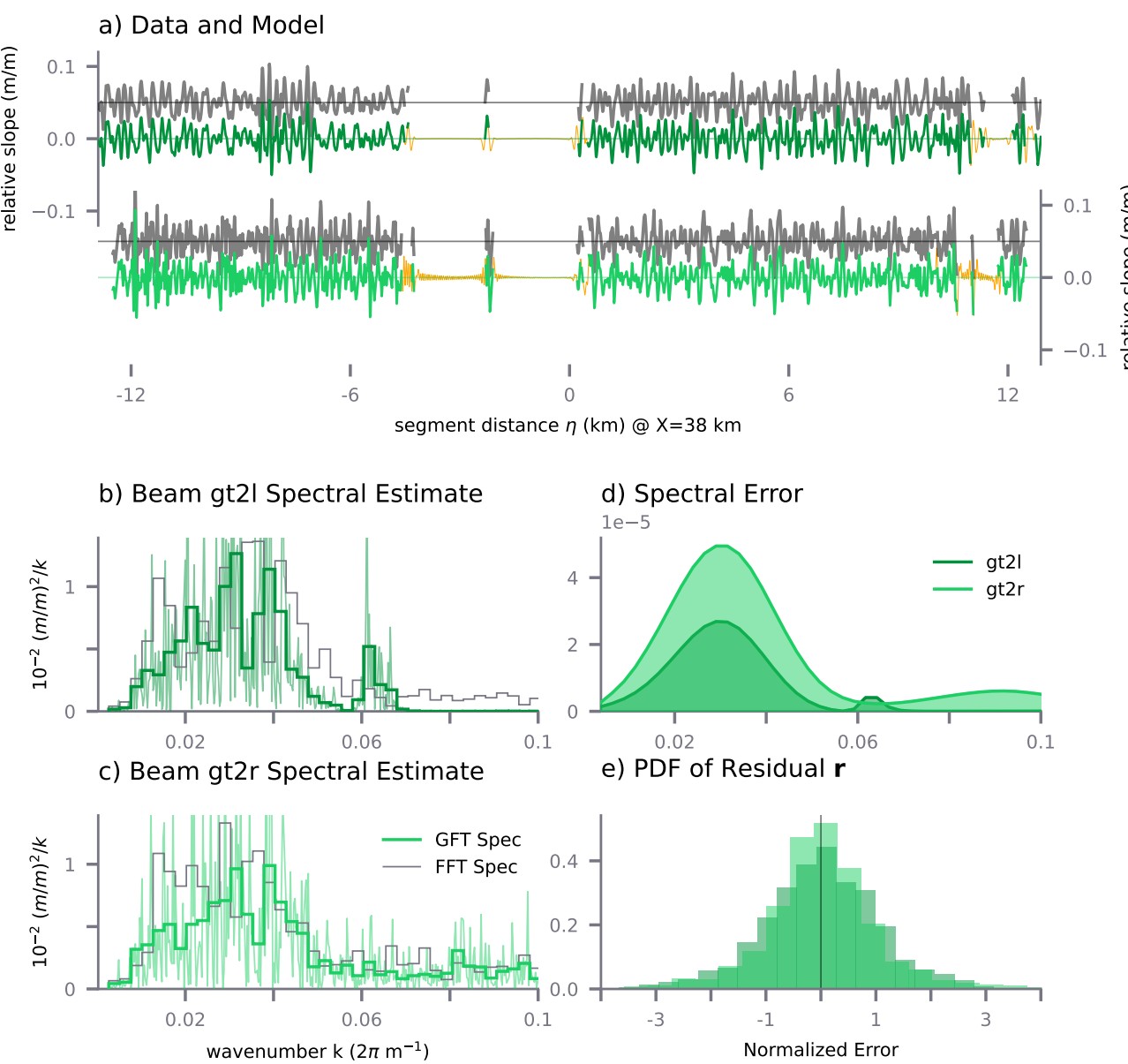

**Figure 3.** Example model and spectral estimate for a 25-km segment in the MIZ for Track 2 (granule `08070210`, see suppl. Table). (a) Data (gray), model (dark and light green), and predicted modeled (orange) photon heights for two neighboring beams, gt2l (dark green) and gt2r (light green). The data is offset by 0.5. (b and c) Corresponding power spectral estimates using GFT (green) and standard DFT (gray). The thick green lines are the GFT estimates re-binned into blocks of $\Delta k = 2.5 \times 10^{-3}$. (d) Power spectral error for both beams as a function of wave number. (e) Residual PDF of **r** for both beams, respectively.

## 2.4 Iterative inversion along each beam

The GFT solution $\hat{\mathbf{b}}$ depends on prior assumptions about the wave spectrum: the model prior $\mathbf{P}$. Since the GFT is iteratively applied along each beam, results from a previous segment inform the subsequent segment as illustrated in Figure 4. Here we describe how a successive application of the GFT along the IS2 beam can lead to an efficient solution assuming that the wave's spectral shape only slowly varies between the segments.

The iterative solution along each beam is initialized at the most ocean-ward edge of the data with a prior $\mathbf{P}_{init}$ that follows a common shape of a narrow banded surface wave field; the Pierson-Moskowitz (PM) spectral slope function (based on Pierson and Moskowitz, 1964). For this segment alone, the PM function is fit to a DFT of the data, and, for the DFT only, any locations with missing data are defined with a slope of zero. This gap-filling creates artificial ringing in the DFT but is sufficient to estimate the spectrum's peak wave number and energy. The PM-spectrum has, in its simplest form, only two free parameters, the peak frequency, and spectral amplitude, which are fit to the DFT power spectra via an objective function that is regularized by the observed spectral peak of the smoothed data (Appendix A, Hell et al., 2019).

The initial inversion of the most equatorward segment $X_0$ is performed using $\mathbf{P}_0 = \mathbf{P}_{init}$ in eq. (6), leading to model parameters $\hat{\mathbf{p}}_0$. For this first segment, a second inversion is applied on the same data, using an updated prior that is a smoothed version of $\hat{\mathbf{p}}_0$ (Fig. 4, left). The smoothing uses a Lanzos running smoother in wavenumber with a stencil-width of $\pm 0.19$ $2\pi\mathrm{m}^{-1}$, or 150 data points. Inversions of the successive segments $X_1, X_2, X_3, ...$ are then performed once, with the prior $\mathbf{P_i}$ a smoothed version of $\hat{\mathbf{p}}_{i-1}$. If missing data does not allow for a successful inversion of a segment, the algorithm is re-initiated as done to obtain $\mathbf{P}_0$ and $\hat{\mathbf{p}}_0$.

This two-stage inversion for segments with no preceding along-track segment ensures that ordinary wave spectra will be identified at the margin while still having the flexibility to allow for more complex wave signals. The effect of the PM prior and details about the derivation is shown in Appendix A.

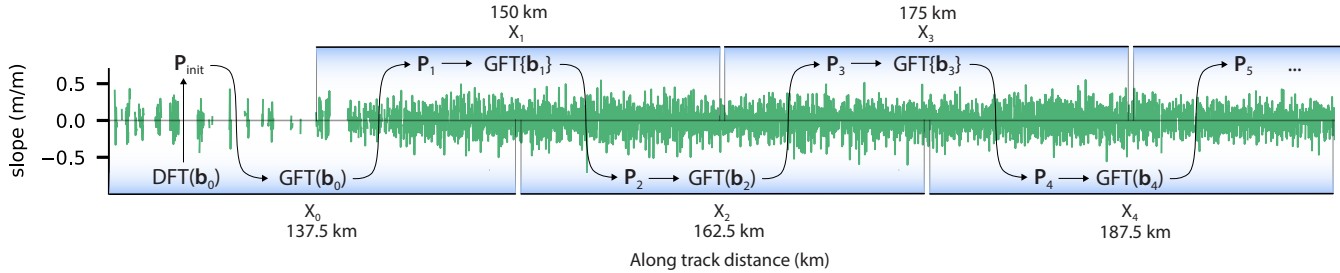

**Figure 4.** Schematic of the harmonic inversion algorithm along an ICESat-2 beam, shown over the surface slope field of a typical beam in the MIZ. The DFT in each successive segment $X_i$ has a prior $\mathbf{P}_i$ derived from the previous segment centered around $X_{i-1}$. For those segments with no prior information available, the prior $\mathbf{P}_{init}$ is generated via a Discrete Fourier Transform (DFT) over the segments centered at $X_i$.

## 2.5 Tracking of wave energy through Sea Ice

The GFT is applied to each 25 km segment with more than 250 data points, leading to wave spectral estimates along each beam. In Fig. 5(a-f) we show wavenumber spectra for each segment and for each of the beams of Track 3 in the Southern Hemisphere on May 2, 2019. The per-segment cross-beam mean (Fig. 5h) and mean spectral error (Fig. 5i) are derived by weighting each segment by its photon density before taking the mean. We define this weighted mean and error as our best estimate of the along-track spectral evolution of wave energy.

The example track shows an attenuating swell signal starting at $X = 0$ km ($X$ is the distance from the ice edge as defined in section 2.1) and a second wave-energy maximum with shorter wavelength ($k' \approx 100m$) at about $X = 150$ km in both, the best estimate as well as in individual beams (Fig. 5a to f, and h). (These wave events are further analyzed in section 4). The corresponding errors are often larger where the wave signals are larger (Fig. 5 i). Instances with a low photon density and more frequent data gaps may fail to invert for the wave signal, resulting in a spectrogram that may not follow expected spectral shapes. These can be identified through their substantially larger error (suppl. Fig. S4 g to i).

The estimated wave numbers are the observed along-track wave numbers $k'$, which are different than the true wave number length $|\mathbf{k}|$ along the incident wave vector $\mathbf{k}$ (see Fig. 1). To estimate a wavenumber spectrum along the dominant propagation direction rather than the direction it is observed, we outline a method to correct this bias in the following section.

## 3 Two-dimensional wave spectra from nearly one-directional observations

### 3.1 Metropolitan estimates of the incident angle

The observed wave spectra are distorted by a misalignment between the wave's incident angle and the beam's direction (Fig. 1c). While the ICESat-2 track orientations are well known, the surface waves can originate from any direction, and the angle between these two directions is $\theta$. If $\theta$ is known, the observed wave number $k'$, or wavelength $\lambda'$ along the beam can be corrected using $k = k'cos(\theta)^{-1}$. The same geometrical distortion will also affect estimates of the attenuation rate between $X_i$-positions along the track (Fig. 1b) because the wave energy attenuates along their dominant propagation direction and not along the direction they are observed by the satellite.

We use the phase lag between weak-strong beam pairs to estimate $\theta$ from the photon data. This requires that wave crests observed in one beam are also observed in another. Using the phase lag to measure the incident angle has several limitations that have to be taken into account when designing an optimization method:

- The phase lag between beam pairs can only usefully be calculated for not too oblique angles (Suppl. Fig. S5, and Yu et al., 2021) and high enough photon densities in both beams. The angle limits in which the phase lag can be resolved depend on the chosen wavelength and the distance between the beams. Since both, wavelength and distance, change for each segment, we here limited the analysis to angles of $\pm 75°$.

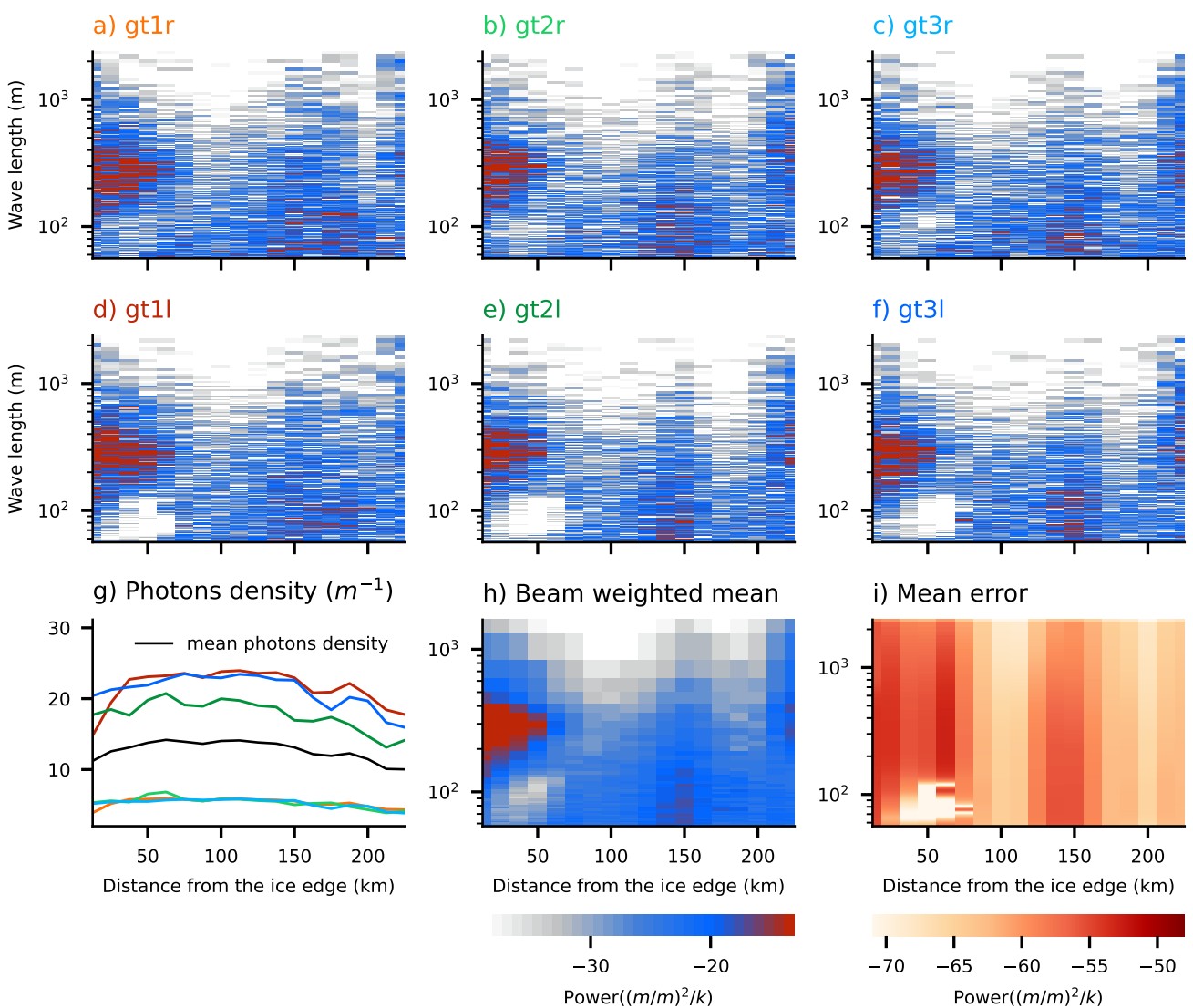

**Figure 5.** Slope spectra for the six beams (a to f) of example Track 3 (granule `05160312`, see suppl. Table) resulting from the GFT inversion (sec. 2.4) for granule 05160312. The photon density per segment (g) is used to calculate the weighted mean spectrum (re-binned as in Fig. 3b,c) (h) and error (i) for each track segment. The horizontal axis is defined as the distance to the most equatorward point with a photon rate exceeding 0.02 photons per meter (sec. 2.1).

– Across-beam phase lag relies on accurately measuring the distance between the beams. Uncertainty in the intra-beam distance $d$ can further obscure angle estimates based on the phase lag (Fig. 1b). Since $d$ varies along the track (suppl. Fig. S6), using the nominal distance rather than the observed value will also bias angle estimates.

- The complex generation and propagation history of waves (Kitaigorodskii, 1962; Villas Bôas and Young, 2020; Marechal and Ardhuin, 2021; Hell et al., 2021a) leads to a dynamic distortion in the incident angle. While a monochromatic plane wave would be coherent across beam pairs, estimating its direction is limited by the periodicity of the waves and observational noise. In reality, the incident swell wave energy at any given time is contained in several wavenumbers although concentrated in a narrow-banded 2D-spectrum (Longuet-Higgins and Deacon, 1957). A narrow-banded 2D swell field leads to wave groups in real space and limits the observable phase lag between strong/weak beam pairs. While a narrower 2D spectrum results in spatially larger coherent wave crests, a broader spectrum spans more random-phase waves, diminishing the observable phase lag between wave crests (suppl. Fig. S7).

With IS2, the incident wave energy along $\mathbf{k}$ is observed along the beam direction for wavenumbers $k' = |\mathbf{k}| \cos \theta^{-1}$. In the case of IS2 beam pairs, we know neither $\theta$ nor the bandwidth of the incident spectrum (Longuet-Higgins, 1984) and both factors limit the possible angle inversion based on beam pairs alone (suppl. Fig. S7). Despite these limitations, we describe in the following how the incidence angle $\theta$ can still be retrieved within these limitations.

As explained in section 2.2, the surface wave field can be interpreted as the superposition of monochromatic plane waves. For a narrow-banded swell spectrum, the majority of the wave energy is contained in a few wavenumbers and hence a superposition of these most energetic monochromatic waves explains most of the surface slope variability. In the following, we optimize the incident angle and phase of the most energetic monochromatic wave numbers using a Metropolis-Hastings (MH) algorithm (Foreman-Mackey et al., 2013). We accumulate the marginal distributions of possible incident angles across the dominant wave numbers and beam pairs, which results in the best guess of possible incident angle. This approach leads to directional wave information similar to the maximum entropy method used in wave buoys (Lygre and Krogstad, 1986).

We focus on the 25 most-energetic wavenumbers of each beam pair and segment $X_i$ based on the GFT result (sec. 2.2). To identify these wavenumbers, the beam-pairs mean wave power is smoothed using a three-wavenumber running mean to select possible wave numbers within the distorted narrow-banded spectrum (similar to the thick green lines in Fig. 3 b,c). For each of these $n = 25$ observed wavenumbers $k'_n$, we define a monochromatic model,

$$\hat{\mathbf{h}}_n(\boldsymbol{\eta}, \boldsymbol{\nu} \,|\, k', \theta, \phi) = \cos(k' \boldsymbol{\eta} + l' \boldsymbol{\nu} + \phi), \tag{11}$$

in the local reference system of the segment centered around $X_i$ and $y = 0$ such that the local along-track coordinate is $\boldsymbol{\eta} = \mathbf{x} - X_i$ and the across-track coordinate is $\boldsymbol{\nu} = \mathbf{y}$ with the observable across-track wavenumber $l' = k' \tan(\theta)$, and the phase $\phi$.

The monochromatic model is then used to define the objective function $\Phi_n$ for each wavenumber

$$\Phi_n = ||\mathbf{b} - \hat{\mathbf{h}}_n||^2 + \beta_\theta \, P_{\theta,n}(\theta), \tag{12}$$

where $\mathbf{b}$ is the normalized slope data of the beam pair, $\beta_\theta$ is a hyper-parameter which controls the regularization $P_{\theta,n}$ of the incidence angle $\theta$ for the $n$-th wavenumber, and $P_{\theta,n}(\theta, k)$ is a prior estimate that we describe in sec. 3.1.2.

The log-probability of the objective function eq. (12) is sampled for each beam pair, selected wavenumber, and along-track position $X_i$. To derive independent estimates of the incident angle for each $n$-th wavenumber we use a MH Scheme (Marcov-Chain Monte Carlo, MCMC, Foreman-Mackey et al., 2013) by first initializing equally-spaced samples of the objective function over the domain $\theta = [-0.42\pi, 0.42\pi]$ and $\phi = [0, 2\pi)$ and advancing the ensemble of samples (ensemble of walkers) using MCMC. The MCMC method will quickly cluster walkers in the areas of low-cost, or small objective function (Fig. 6b, black dots). A high density of walker positions is then interpreted as a high likelihood of an incident angle and phase for the chosen wavenumber (Fig. 6b, black dots).

We derive a sample of the joint phase and angle distribution by advancing the walkers 300 iterations, and only the last 270 iterations for each walker are used to establish the joint histogram $D(\theta, \phi)$. The joint histogram $D$ is then marginalized for the incident angle $\theta$ and normalized to a probability distribution function (Fig. 6c). This procedure is repeated for each selected wavenumber $k_n$ and for each (available) beam pair per segment $X_i$ (suppl. Fig. S8 a to c). The best incident angle PDF $\theta_{PDF}(X, k, beampair)$ can then be derived using the weighted average across wavenumber, beam-pair, or both.

An example of the resulting beam- and wavenumber-average PDF is shown in figure 7b for Track 3 at $X_i = 87$. Here, the individual PDFs are weighted by the mean power of the respective wavenumber and the number of data points per segment pair. The most likely incident angle is at $-37°$, while two other angles $-63°$ and $0°$ also show high likelihood. Marginal PDFs with multiple maxima are a typical result for this method and appear in many other tested sections and tracks (not shown). They come from different maxima in the joint PDFs of different wavenumbers. If one trusts the angle estimate of a single wavenumber, this result can be interpreted as a wave field with waves from multiple directions. The alternative – likely better – perspective is that the marginal PDF of a single wavenumber is not a robust estimate of the incident angle, and hence the PDF in figure 7b helps estimate the uncertainty of the method.

### 3.1.1   Robustness of the Marginal PDFs

The limitations in retrieving the incident angle (sec. 3.1) lead to a low signal-to-noise regime and demand a careful evaluation of the method for sampling the objective function $\Phi_n$. While larger samples may ensure convergence of the distribution estimate, a large sampler of each wavenumber, beam pair, and segment may not be necessary or computationally affordable. We decide for a systematic under-sampling of each realization of the marginal PDFs $\theta(X_i, k, beampair)$ and, in a second step, make a super-sample from the marginal PDFs across beam pairs and wavenumber if needed.

When combining the systematic under-sampled PDFs, their super-sample will still provide a good estimate of the mean incident angle and its standard deviation. Each realization of the joint $\theta$-$\phi$ distribution requires 6750 function evaluations for 270 iterations per walker. The walker's auto-correlation is about 20 to 30 iterations, which implies that each joint distribution maybe not be well established (the effective degrees of freedom per walker are about 9 to 14). Hence the marginalization of each joint distribution may misrepresent the angle uncertainty (i.e. a too wide distribution of the walker's PDF). To reduce uncertainty, we take a super-sample of the marginal PDFs, by averaging across wavenumbers and/or the three beam pairs. The super-sampling results in a statistically robust result with $5 \times 10^5$ function evaluations per segment $X_i$, which is about $3.5 - 5.5 \times 10^3$ effective degrees of freedom per segment. Longer Markov Chains, i.e. more iterations, may result in a better

sampling of the individual fit, but may not affect the overall result since they sample from generally smooth objective functions in a low signal-to-noise regime. However, in cases where a directional estimate per wavenumber or beam group is needed, the MCMC iteration length can be adjusted.

### 3.1.2 Constraining direction estimates with other data products

A sampling of the objective function eq. (12) as described in sec. 3.1 results in a joint distribution of most likely incident angles and phases per sampled wavenumber $k_n$. This joint distribution may have multiple equally likely maxima, i.e. multiple likely incident angles due to the periodicity of the wave ($2\pi$ ambiguity). As illustrated in figure 6d (shading) this can lead to a) maxima for positive and negative incident angles and b) multiple maxima on both sides. To break the symmetry in the marginalized PDF of incident angles (Fig. 6e) we define a prior $P_\theta(\theta, k)$ in the objective function using ridge-regression (Appendix B). The effect of the prior on the joint- and marginalized distribution is shown by comparing Figure 6 b and c with d and e. Here we inform the prior with WW3 global hindcast wave-partitions (Tolman, 2009, using the Integrated Ocean Waves for Geophysical and other Applications (IOWAGA) hindcast)). WW3 must be treated with caution due to wind-observational biases in the Southern Ocean (Belmonte Rivas and Stoffelen, 2019; Hell et al., 2020). This wave hindcast is currently the only readily available dataset for this global purpose, and priors from observational datasets would improve the quality of this data and the overall wave inversion.

The level of certainty in the WW3 prior is expressed in the hyperparameter $\beta_\theta$ and the performance of the MCMC sampling is sensitive to its value (eq. 12). Since validation of the WW3 prior is limited, we set $\beta_\theta = 2$. Its effect on the objective function can be seen by comparing the shading in figure 6 b and d. The choice of $\beta_\theta = 2$ leads to the desired result in breaking the directional ambiguity while not fully determining the incident angle distribution (Fig. 7a). We tested other values of $\beta_\theta$ but found empirically that higher values tend to overfit to the prior, and lower values do not break the ambiguity well.

This method is limited to angles of about $\pm 75°$ deviation from the nominal track direction. More oblique, i.e. steeper incidence angle can not be captured by this method because a steeper angle requires more coherence between wave crests. The coherence of a single wave crest is, however, limited by the curvature of the wave spectrum and not well known (suppl. Fig. S7). In addition, the model has a $180°$ ambiguity such that waves coming from the equator side of the track (as assumed), or waves coming from the pole side (less likely) can result in the same phase lag and hence in the same incident angle, even though they come from the opposite direction.

## 3.2 Two-dimensional spectra in along-track data

With the spectral and angle estimates (sec. 2.2 and 3.1), we now can describe waves observed along-track in terms of their two-dimensional wavenumber spectra (Fig. 8). The estimated wavenumber amplitudes $\hat{\mathbf{b}}$ (eq. 2) are corrected by $\cos(\theta)^{-1}$ using the most likely incident angle (sec. 3.1, Fig. 7b) resulting in the corrected wavenumber spectrogram (Fig. 8a). We use the most likely angle along the track, although the above analysis can provide angle distributions for each segment $X_i$ and wavenumber $k$ (sec. 3.1.2, Fig. 8b).

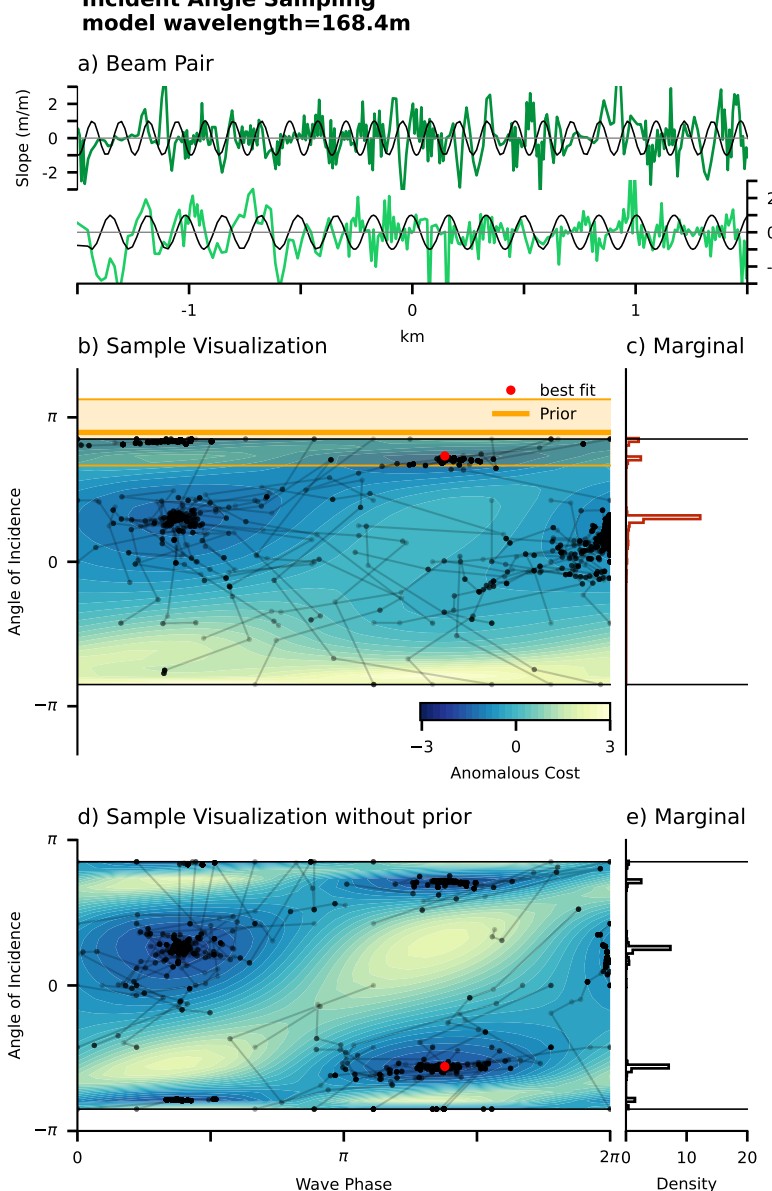

**Figure 6.** MCMC angle estimate using a monochromatic wave model with and without a prior $P_\theta$. (a) Data of the beam pair gt1l and gt1r (light and dark green) and model (black, eq. 11) for a segment of Track 3 (granule `05160312`, centered at $X_i = 62.5$ km. (b). The objective function with prior $P_\theta$ is sampled using a brute-force method (shading) and MCMC (black dots and lines). The prior angle $\theta_0$ and prior uncertainty $\sigma_\theta$ for this wavelength (168.4 meters) are shown as thick orange lines and shading. The best fit using a dual-annealing method is shown as red dot (Tsallis and Stariolo, 1996). (c) Marginal PDF of the incident angle $\theta$ from MCMC sampling. (d) and (e) same as (b) and (c) but without the prior $P_\theta$.

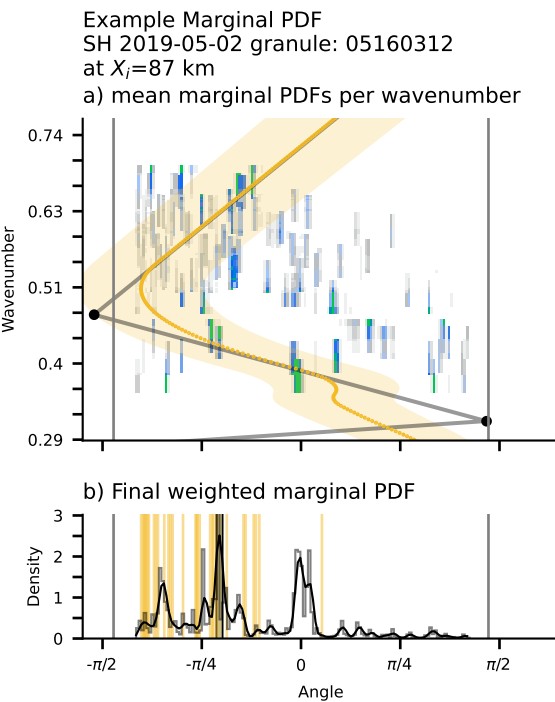

**Figure 7.** Influence of the WW3 prior on the marginal PDFs of Track 3 (granule `05160312`) at $X_i = 87$ km. (a) Mean marginal PDFs for all beam pairs as a function of wavenumber. The WW3 spectral partitions are shown as black dots and the interpolated prior $\theta_{0,i}$ and its spread $\sigma_{\theta,i}$ (eq. B1) are shown as orange line and orange shading. (b) The smoothed weighted mean across the pairs of the most likely incident angle is shown as a black thick line and the WW3 priors for all wave numbers are used as orange lines.

The corrected power spectrum and directional distribution (Fig. 8 a,b) can be expressed as directional surface wave spectra every 12.5 km in the MIZ, similar to conventional surface wave buoys (Fig. 8 c to e). This permits tracking the attenuation of wave energy per frequency in MIZ. In the case of Track 3 (granule `05160312`), for example, we see a wave event coming from about 45° to the right of the ground track that mostly attenuates in the first 75 km from the sea ice edge, while the overall attenuation rate is similar between the six beams (Fig. 5a-f). One could identify a migration of the peak wavelength from about 275 meters to about 300 meters within 12.5 km (Fig. 8 d,e, similar to Alberello et al., 2022); we leave this analysis of the attenuation to future work.

Past the primary wave event, a second signal further into the ice with energy on scales shorter than 200 meters extend from $X > 75$ km about 100 km or more (Fig. 8a,b). This signal is at shorter wavelengths than the identified cut-off frequencies for the event at the track beginning. Without further information about the ice conditions, we suggest that this short-wavelength energy deeper in the sea ice is due to sea ice variability itself rather than due to waves.

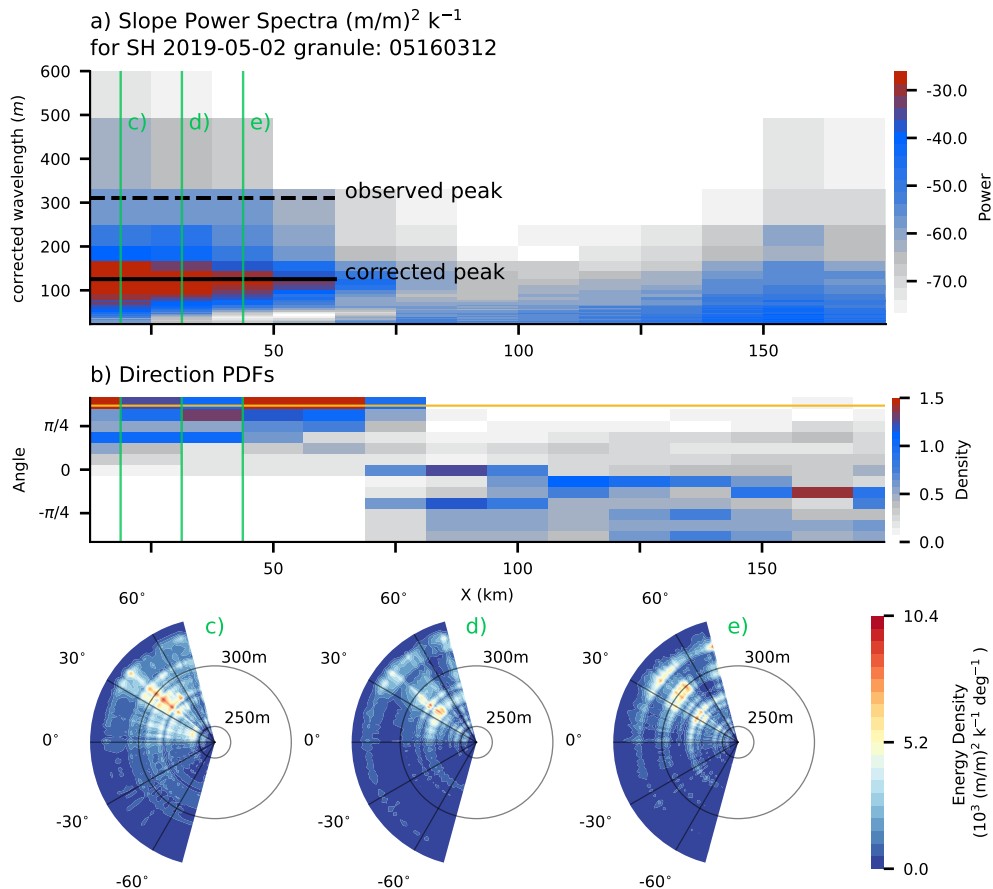

**Figure 8.** Final estimate of the slope power (a), directional (b), and joint (c,d,e) surface slope spectra every 12.5 km along an ICESat-2 beam (example Track 3). (a) The angle-corrected, cross-beam average spectrogram as a function of estimated wavelength $\lambda$ and distance from the ice-edge $X$. The spectrogram is re-binned in $2.5 \times 10^{-3}$ wavenumber segments and its wavelength is corrected by about a factor of three due to a peak incident angle of about $66°$ (orange line in b). The observed and corrected peak wavelengths $\lambda'$ and $\lambda$ are shown as dashed and solid black lines. (b) Mean directional PDF between $\pm85°$ every 12.5 km, rebinned into $10°$ segments. (c) to (e) Rolling-mean smoothed directional wave spectra at the positions indicated at the respective green lines in (a) and (b).

## 4 Isolating the wave signal in the along-track data

The estimate of the wave signal from sec. 2.3 can be used to decompose wave and ice surface variability. Each photon retrieval is a super-position of ocean waves and sea ice signals like surface roughness, floe size, and freeboard height. A decomposition of the surface variability between waves and ice can rely on the coherence of across-beam wave statistics, a common noise level in wavenumber space, and an approximate scale separation of the dominant wave energy and the sea ice. For the purpose of decomposing the data we define similar signal-to-noise levels across beams in each 25 km segment (Fig. 5 a to f).

The results of the GFT (sec. 2.3) are used to delineate ATL03 photon heights between wave and sea ice surface variability. We construct a binned wave height field along the track from the GFT-derived surface spectrum, by filtering out high-wavenumber

components that likely do not correspond to swell waves. In Figure 9 we show the identified low-pass filters and the displacement spectrum (m$^2$ k$^{-1}$) rather than the slope spectrum ((m/m)$^2$ k$^{-1}$), as in Fig. 8) to better separate the high-frequency noise from the lower-frequency waves. The low-pass filter is defined by a cutoff-wavenumber $k'_c$, the first wavenumber where the observed power spectrum changes slope. A change in the slope of the displacement spectrum in log-log scaling from the expected slope of surface wave spectra ($k^{-5/2}$ or similar, Toba, 1973) to horizontal indicates a change in the signal-to-noise regime in the data (Fig. 9). Hence, horizontal slopes at high wavenumber indicate Gaussian (white) noise, while steeper slopes at lower wavenumber result from wave-wave interaction (Kitaigorodskii, 1962; Hasselmann et al., 1973). The critical wavenumber $k'_c$ between both regimes is found using piecewise regression on the weighted cross-beam log-log power spectrum (Fig. 9, Pilgrim, 2021). In cases where the piecewise regression fails to identify a separation a steep and horizontal slope, no primary wave spectrum is identified and no low-pass filter is applied (Fig. 8b, $X > 87.5$km).

For illustrative purposes, we define a low-pass filter by setting $k'_c$ as the cut-off wavenumber. This low-pass filter potentially creates artificial ringing in real space and for better results, this should be replaced by a more complex filter design. Here, wavenumbers higher than $k'_c$ are excluded from the wave height model of all beams (Fig. 10a, gray and blue lines and the gray area in the inlet) by truncating the wavenumber space of the slope model $\mathbf{p}$. From this truncated slope model, we can directly construct a coefficient matrix for the wave-height model $\hat{\mathbf{z}}$ for each beam by integrating in wavenumber space. The coefficient matrix of the wave-height model $\hat{\mathbf{z}}$ is

$$\hat{\mathbf{d}} = [-c_1, -c_2, ..., -c_c, a_1, ...a_c,]^T,$$

where $c_c$ and $a_c$ are the model amplitudes corresponding to the cutoff frequency $k'_c$ (note the integration of the trig-formula changes order and sign of the indices). The reconstructed wave-height model $\hat{\mathbf{z}}$ can then be directly calculated from the original regressor matrix,

$$\hat{\mathbf{z}} = \mathbf{H}\,\hat{\mathbf{d}}\,\mathbf{k'_c}^{-1}, \tag{13}$$

with $\mathbf{k'_c} = [k'_1, k'_2, ..., k'_c, k'_1, ..., k'_c]^T$ as shown in figure 10b blue line. The residual between the height model $\hat{\mathbf{z}}$ and the observed smoothed photon heights $\mathbf{z}$, $\mathbf{z}_{free} = \mathbf{z} - \hat{\mathbf{z}}$, is an estimate of the freeboard height in absence of the influence of waves (Fig. 10d). The residual $\mathbf{z}_{free}$ has similar data density to the original ATL03 photon retrievals but may reveal secondary, non-wavelike structures in the photon heights as shown in figure 10d. We provide additional examples in suppl. Fig. S9 and S10.

Decomposing heights into wave and sea-ice components allows us to estimate the fraction of the total height variance that is neither due to waves nor photon variance on scales shorter than the 10-meter stencil. As shown in figure 10e, the majority of the total variance is due to the photon variance around its 20-meter stencil mean for scales smaller than the stencil (Fig. 10d red line and black dots, Kwok et al., 2021). In this particular track, wave variance comprises then another 20% to 50% of total photon height variance. The remaining variance, about 5% to 20%, is then neither due to waves nor the photon cloud. It is from differences in the observed and modeled wave heights, $\hat{\mathbf{z}}$ and $\mathbf{z}$, and we assign this to sea-ice-related variability.

The distribution of the residual statistics is, by model design, approximately Gaussian (Fig. 3e) and hence non-wave signals with non-linear imprint could contaminate the wave estimate and decomposition. If the contribution of non-wave signals to the

across beam average is minor, this decomposition removes waves as the dominant source of variance on scales larger than 20 meters. This allows for additional analysis of the residual signal, and more consistent surface height signals in wave-affected and low-sea ice regions. A better filter design can further improve this separation between waves and sea ice.

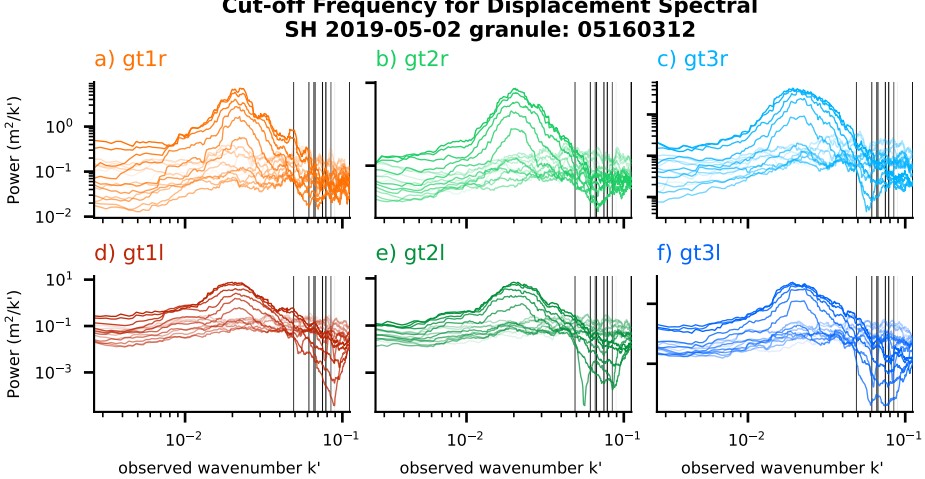

**Figure 9.** Sections of derived displacement power spectra $\hat{\tilde{S}}_h$ for Track 3 (Fig. 8) for each beam (a to f). The black lines show the position of the estimated cut-off wavenumber $k'_c$ based on piece-wise regression on the weighted mean of all beams. Darker colored lines show sections closer to the sea ice edge and lighter colors show sections further into the sea ice. Estimates of $k'_c$ at $X > 100$ km fail due to no slope separation (compare to Fig. 8a). Note that we show the uncorrected displacement spectrum rather than the corrected slope spectrum as in figure 8. The corrected spectral peak at $\lambda = 125$ meter (Fig. 8) corresponds to the peak about $k' = 0.02$ meter$^{-1}$ in this figure.

## 5 Discussion

ICESat-2 photon data frequently shows wave-like signals in sea ice and these substantially impact the marginal ice zone. In this paper, we show, for the first time, how paired laser-altimeter observations can be converted into directional surface wave spectra. We describe a two-part algorithm that efficiently decomposes the IS2 photon retrievals into a surface wave signal as well as variability due to sea ice. The first (GFT) part of the algorithm is based on a linear inversion method to fit wavenumber coefficients to the ATL03 data (sec. 2.2 to 2.4), and the second (MH) part uses a non-linear inversion method that optimizes for most-likely wave incident angles (sec. 3.1).

The combined method provides a highly-resolved 2D-surface wave spectra every 12.5 km along each IS2 track (Fig. 8a,b) as well as an improved surface height estimate when the wave signal is removed (Fig. 10). The surface wave estimate relies on the one hand on the redundancy across beams to optimize the signal-to-noise ratio in wavenumber space and on the other hand on the difference across beams for the angle inversion. The iterative solution proposed here leads to an interpretation of the IS2 track as a streak of two-dimensional wave spectra, including error estimates on each variable (Fig. 8c to e).

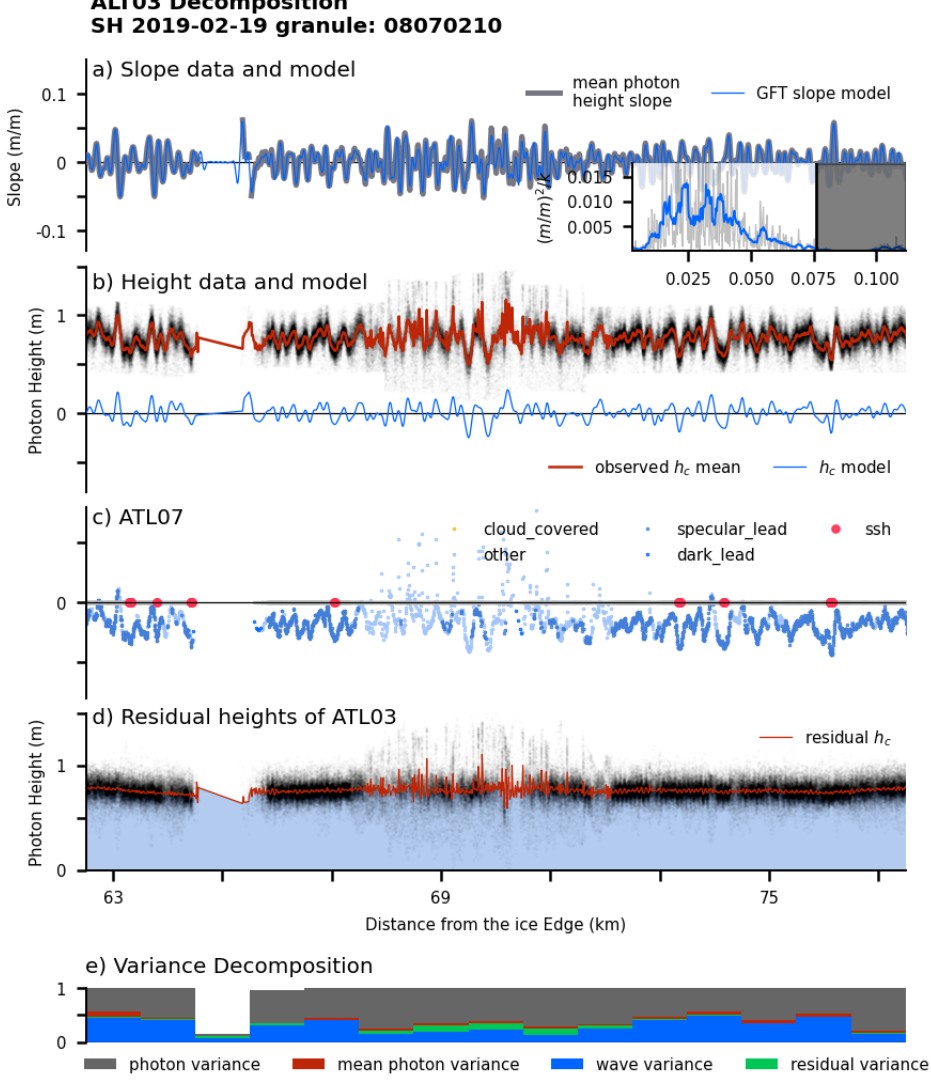

**Figure 10.** Variance decomposition of the photon-cloud data (ATL03) based on the GFT of example Track 2. (a) The observed mean surface height slope $\mathbf{b}$ is shown in gray and the truncated model $\hat{b}$ in blue. The inset shows the corresponding smoothed spectral amplitudes and cut-off wavenumber $k_c'$ as a black line. (b) The observed photon cloud is shown as black dots, re-binned data $\mathbf{z}$ as a red line (sec. 2.1), and the reconstructed surface heights as a thin blue line $\hat{\mathbf{z}}$ (eq. 13). (c) Corresponding ATL07 surface heights product shown as in figure 2. (d) Residual photon heights (black) and binned heights (red) using $\hat{\mathbf{z}} - \mathbf{z}$. (e) Variance fraction every kilometer with the fraction due to the photon cloud (gray), the truncated wave model $\hat{\mathbf{z}}$ (blue), the variance of wavenumbers $> k_c'$ (green), and the residual of the model $\mathbf{r}$ (red, eq. 1).

We identify the range of wavenumbers that contain wave energy in each segment by establishing a dynamic noise level (sec. 4). Removing the wave energy as a dominant source of variance reveals additional structure in the ATL03 photon cloud data that is not as readily present in the ATL07, or other higher level products (Fig. 10d, or suppl. Fig. S9 and S10), either because it is obscured by wave signals or the data is not present. Even though we do not investigate the residual photon heights

further, we believe that a removal of the wave signal may have substantial benefits for understanding the sea ice structure and classifying photon data for ATL07 products and above. Not removing the wave signal likely leads to an aliasing effect of the waves into the freeboard height (compare panels b and d in figure 10).

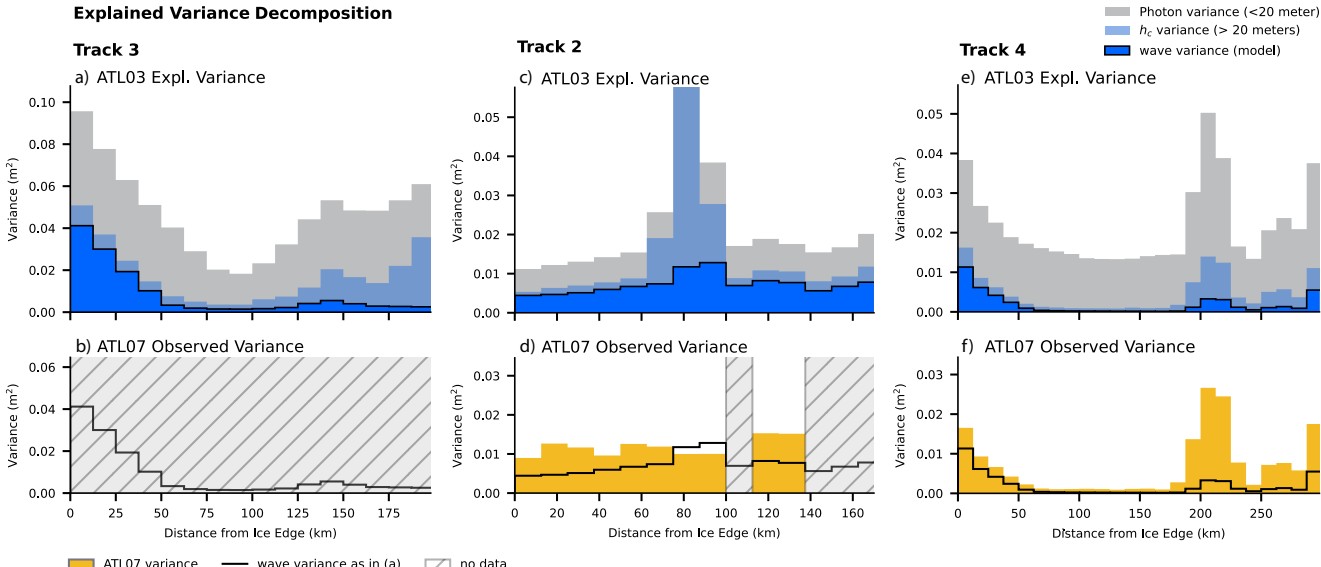

**Figure 11.** Variance decomposition of ATL03 photon cloud data and comparison with ATL07 data. (a) Across-beam averaged variances for Track 3 (granule `05160312`, see suppl. Table). The photon cloud variance is shown in gray (same as Fig. 10b black dots), the variance of the 20-meter stencils is shown in light blue (same as Fig. 10b red line), and the variance of the low-pass filtered wave model in dark blue with a black outline (same as Fig. 10b blue line). (b) Variance of ATL07 based on the provided segment heights. Gray hatched areas indicate no data in ATL07 for this track. The black line is repeated from panel (a) for comparison. (c and d) Same as (a) and (b) but for Track 2 (granule `08070210`). (e and f) Same as (a) and (b) but for Track 4 (granule `05180312`)

Our quantification of wave energy allows for an improved understanding of the observed surface elevations in sea-ice-covered areas. We showed that ocean surface waves have an important contribution to the variance in the MIZ. While the example tracks have a substantial amount of variance in photon height on scales smaller than 20-meters (Fig. 11a,c,e gray area), the variance on scales longer than 20-meters is clearly dominated by the effect of waves (Fig. 11a,c,e blue area). Especially in the MIZ, only a small fraction of this re-binned variance is due to non-wave-like features of the surface (Fig. 11a and e gray-blue area at 0 - 50 km from the ice edge).

The chosen examples show a clear wave signal that can be separated from high-frequency noise by using a simple low-pass filter with a cut-off frequency $k'_c$ (sec. 4). This cut-off frequency assumes a scale separation between waves and sea ice variance, that generally may not exist. The identified cut-off frequency lies in the plausible range of wind waves ($k = 0.05$ to $0.1$). In cases with wind waves and complex sea ice structures (Alberello et al., 2022), a separation between the wave and sea ice signal is hampered. Without adding observations other than ICESat-2, we are not able to directly differentiate between wind sea and well.

This wave-induced variance in the photon cloud of ATL03 can, under certain conditions, also be captured by ATL07. However, we suggest that the ATL07 algorithm also can potentially capture an aliased wave signal, or can fail to provide a sufficient sea-ice height product at all. While in the case of Track 3 (Fig. 11a,b), waves clearly affect the observation, ATL07 does not provide data in this region (see also suppl. Fig. S1) and so our inversion method using ATL03 allows for improved MIZ freeboard data. In other cases, like Track 2 (Fig. 11d), ATL07 does provide data at some but not all locations along the track, even though there is a high photon density throughout the track.

In the MIZ, ATL07 variance can exceed the variance estimates of waves, indicating potential aliasing of wave-induced signal to other scales (Fig. 11d). This aliasing can be due to the binning of data based on photon counts which result in varying bin length. Varying bins-length potentially sub-samples the wave's energy at scales on, or around the Nyquist frequency of the dominant waves. Since both the wavelength and photon density highly vary, it is generally unknown whether or not the photocount-based measure samples these wavelengths correctly and does not negatively affect freeboard retrievals or wave energy estimates in the MIZ .

## 5.1 Applying the Generalized Fourier Transform method for along-track wave spectra

We chose a generalized Fourier transform (GFT) method for the wave field inversion instead of more common methods like DFT, or Lomb-Scargle (LS, sec. 2.2, Lomb, 1976; Scargle, 1982; Wunsch, 1996; Kachelein et al., 2022). In contrast to the DFT and LS, the GFT is variance conserving method that can be applied to unstructured data and does not require periodicity over an (arbitrary) window length.

As with any linear inverse method, the GFT assumes Gaussian statistics, which is obeyed by linear waves but potentially violated by sea ice surface variability. To minimize the effect of sea ice heights on the wave inversion, we use mean surface slopes rather than heights (sec. 2.1), which results in an approximate Gaussian residual (Fig. 3e).

The GFT is customizable to the wavenumbers of interest and additionally provides uncertainty bounds on all parameters. In turn, it comes with the requirement to apriori know what spectral resolution is needed. Given Atlas's high-resolution photon cloud data, we chose to resolve the plausible wavenumber range for surface waves on a resolution about twice the one of the DFT. Other, more targeted, narrow-banded wavenumber spaces are possible, but here we choose wavenumber ranges that are the most general for surface waves. Higher resolution, especially at long wavenumber allows us to provide new insights into how narrow-banded the surface wave field is; a parameter that is related to the surface's curvature and likely important for wave-induced sea-ice breakup (Meylan and Squire, 1994).

A major advantage of the GFT is that it can be extended to inversions of the wave field for each IS2 track by coupling neighboring or overlapping segments, similar to Kalman inversion methods. We illustrate this by simply iterative updating the data segments and models priors (sec. 2.4, Fig. 4). This coupling of inversions along segments advances the task on hand to the task of solving the coupled inversions consistently along each beam, rather than independently for each segment. The same idea can be extended by coupling the model prior $\mathbf{P}$ across beams. This coupled approach ensures smoothly varying wave statistics along and across-track, with the amount of smoothness tuned through the amplitude of $\mathbf{P}$ in (eq. 6). Future comparison with other observations will help to constrain the amplitude of $\mathbf{P}$.

## 5.2 Applying the Metropolis-Hastings method for wave angle inversion

The inversion for the wave's incident angle is based on the cross-beam phase lag (sec. 3). The phase lag between two beams is limited by the geometry (Fig. 1b), a difficult-to-estimate property of the wave field (i.e. its "groupness", suppl. Fig. S7, Longuet-Higgins and Deacon, 1957), as well as observational noise (sec. 3.1). We choose to approach this low signal-to-noise problem with a super-sample of marginal distributions derived from independent MCMC samples of monochromatic plane waves. The unweighted mean of this method across all wavenumbers is similar to the lag-cross correlation of the beam pair. However, by focusing on a limited set of energy-containing wavenumbers, the signal-to-noise ratio improves above a lagged cross-correlation approach. Limiting the sample to the most energetic waves and using a prior raises the signal-to-noise ratio and is what enables an inversion of the approximate wave angle (Fig. 7d).

The quality of MH inversion method depends on the wavelength, wave amplitude, and curvature of the wave spectrum. The longer the wave the better the phase lag can be observed, but those are not the most energetic. In turn, the most energetic waves have typically shorter wavelengths that are of 80-250 times the segment length (25km), which can lead to multiple minima in the optimization due to a $2\pi$ ambiguity. Finally, the curvature of the wave spectrum characterized the length of wave groups, which in 2D, erode the ability to observe the average phase lag between the two beams (2nd bullet point in sec. 3.1, suppl. Fig. S7).

The inversion is limited by the geometry of the observation (Fig. 1c). Waves coming from steep angles relative to the IS2 track cannot be resolved, such that our 2D-wave field estimates are limited in range (about $\pm 75°$). This problem may be overcome by using better, observationally-based priors, or enriching our WW3 priors with data from other sources.

The angle inversion method generally can sample multiple incident angles, i.e multiple minima in the objective function, and may be able to detect multiple wave systems from different directions (Alberello et al., 2022). However, after testing the effect of wave groups and uncorrelated noise on the sampling method, we came to the conclusion that the signal-to-noise ratio is too low for a frequency-dependent angle correction, even after adding a prior. Hence, for the purpose of angle correction, we here assume that all wave energy comes from the same angle, i.e. the angle of the most energetic waves. Even though the true wave field may be a superposition of multiple wave systems with varying directions, the single incident angle is justifiable here, because we are focused on the attenuation and propagation of the dominant wave energy.

The low signal-to-noise of the angle inversion requires regularization (sec. 3.1). Since directional wave observations co-located with IS2 tracks are sparse and not readily available, we relied on Wave Watch III (WW3) hindcast models as a prior (IOWAGA Tolman, 2009). The wave hindcasts may perform sufficiently well in the Northern Hemisphere but are known to have limitations in the Southern Ocean MIZ, potentially due to wind biases (Belmonte Rivas and Stoffelen, 2019; Hell et al., 2020, 2021b). The lack of certainty in WW3's peak direction and frequency is expressed in the value of the hyperparameter $\beta_\theta$ (eq. 12). A value of $\beta_\theta = 2$ leads to the desired behavior of breaking the symmetry (compare shading in figure 6 b and d) but not imposing the optimization result through the prior (Fig. 7d blue and orange lines).

The proposed MCMC method shares aspects with the Wavelet Directional Method (WDM, Donelan et al., 1996, 2015), which decomposes the signals of at least three stationary wave observations into wavelets for each frequency. Similar to our

method, WDM uses the phase lag of the wavelets between the three stations to identify a wave incident angle per frequency. WDM could be applied to transects of the wave surface as present in our analysis. However, ICESat-2 only provides two neighboring laser beams, and other beam pairs are too distant (about 3 km) for coherent phase analysis. In addition, the signal-to-noise may be substantially lower in the ICESat-2 observations, as wave crests are potentially distorted by sea ice structure. Therefore, we introduced a wave-angle prior (eq.12) to break the ambiguity in the observed phase lag (Fig. 6b,d, shading).

## 6 Conclusion

We proposed and tested a method to decompose photon retrievals from the ICESat-2 satellite into a surface wave signal and residual variance. The surface wave signal is identified using a Generalized Fourier Transform, and it can be expressed as a directional surface wave spectrum by adding a Metropolitan-Hastings sampling method to identify the incident angle of the dominant wave energy (sec. 3). The wave and sea ice signal is separated using a simple cut-off frequency that implies a separation of scales.

Surface waves and sea ice can have complex non-linear interactions that are important to model for improving sea ice and climate projections. This method will enable us to observe the interaction between the dominant waves and sea ice by utilizing large datasets from ICESat-2 (IS2). However, I2's nearly one-dimensional snapshots of the surface height can hardly capture all possible interactions between waves and ice. Besides gaps in the observations due to clouds and varying photon densities between sea ice types, a correct wave field inversion is only possible with sufficient data density and a limited range of incident angles (sec. 3). Even though this method outlines a better, more transparent wave-field inversion than a DFT, it remains to be seen how the interaction of those limitations can be used to provide a highly-resolved global wave-in-ice product. Comparisons with other data sources, either from in-situ or remote sensing observations, are needed to understand these limitations better and validate this method.

Waves and sea ice have scales ranging multiple orders of magnitude such that it is challenging to separate both in the IS2 observations. The choice of the parameters in this analysis (10-meter bins, 25km segment length, and the slope-based cut-off frequency $k_c'$) focus on identifying swell wave events routinely created by synoptic storms (Hell et al., 2021a). However, even on these scales (80 to 350 meters), a separation between wave and sea-ice signal may only be possible when the sea ice variance is weak on those scales and the data is not too gappy, as in the chosen example tracks (Fig. 3, Fig. 10). High levels of sea ice variance or frequent data gaps will lead to systematic biases and aliasing effects in the wave spectral estimates. To identify these more complex cases, we proposed mitigation strategies that exploit the fact that swell spectra normally vary over larger scales than the segment length (12.5 km) or the separation distance of the beam pairs (3 km): By applying an iterative inversion (Fig. 4), using cross-beam average (Fig. 5), and provide error estimates in frequency, real space, and direction (eq. 9, eq. 10, Fig. 5i, Fig. 7b), the method provides ample auxiliary data to detect unusual features in the observations. Complex cases that show a large spread between beams, or large errors, can be identified and excluded from further analysis (suppl. Fig. S4).

Despite the shortcomings and limitations of individual track inversions, the method has promise, especially when applied at scale. We expect $10-15\%$ of the IS2 tracks in polar regions to be dominated by waves (Brouwer et al., 2021), which means

there is already large collection of diverse wave observations from IS2. Because IS2 can also record freeboard heights, floe sizes, and sea ice types, this analysis can provide complimentary sea ice information to constrain dynamics in the MIZ. This will be used to statistically constrain parametrizations of wave-attenuation in sea ice (Fig. 5) and can potentially leverage the wave-removed residual signal to improve ice classification (Fig. 10 d).

Finally, while the here developed method is customized for ICESat-2 photon retrievals, this approach applies to any unstructured quasi-instantaneous observation of the ocean surface. In the case of IS2, cross-track information is limited, but other future remote-sensing methods may have complimentary information about the surface wave field. Such an inversion could combine data from IS2, SAR, and CFOSat to help constrain the surface wave filed in the MIZ or the open ocean (Collard et al., 2022).

*Code and data availability.* The algorithms are available through Hell (2022a, DOI: 10.5281/zenodo.6908645) and data are available through Hell (2022b, DOI: 10.5281/zenodo.6928350). The IS2 ATL03 data (Neumann et al., 2021) and ATL07/10 data (Kwok et al., 2021) is available through NSIDC (https://nsidc.org/data/icesat-2/data-sets), or OpenAltimetry (https://openaltimetry.org/data/icesat2/, Khalsa et al., 2020). The wave model data are available through the Integrated Ocean Waves for Geophysical and Other Applications (IOWAGA) project: ftp.ifremer.fr/ifremer/cersat/products/gridded/wavewatch3/hindcast/. The analysis uses and modifies the *icesat2 toolkit* (https://read-icesat-2.readthedocs.io/)

## Appendix A: GFT priors

## A1 Data Prior $\mathbf{R}$

We define the data prior $\mathbf{R}$ based on the surface slope uncertainty, as

$$\mathbf{R} = \beta_R \, \sigma(\mathbf{b})^2 \, \frac{\boldsymbol{\sigma}_h}{\Delta x}$$

where the stencil width is $\Delta x = 20$ meters, $\sigma(\mathbf{b})$ is the standard deviation of the data $\mathbf{b}$ within the segment, $\boldsymbol{\sigma}_h$ is the vector of standard deviation of each data point (each stencil, sec. 2.1), and $\beta_R$ is a tuning parameter that determines the ratio of the model and data priors in eq. 6 (Fig. 2 blue lines). The standard deviation, or error, of the data is divided by the stencil size to get an error in units of surface slope, and the variance of the data is then used to amplify the prior $\mathbf{R}$ to scale it against the model prior.

To avoid over-fitting the ratio of the model prior $\mathbf{P}$ and data prior $\mathbf{R}$ has to be adjusted. For this we try different values of $\beta_R$ and set it to $10^2$ such that the distribution of the residual $\mathbf{r}$ is approximately Gaussian and that

$$||\mathbf{r}|| = ||\mathbf{b} - \mathbf{H} \, \hat{\mathbf{p}}|| \approx 1,$$

as shown in Figure 3e. At locations with no data, this results in a model decay to zero (Fig. 3 orange lines) on scales similar to the data's auto correlation.

## A2   Model Prior P

As described in section 2.4, the GFT's prior for initial segments are derived from a PM-spectrum that is fitted to a DFT of the segment data (Fig. A1b gray and black-dashed line). The initial prior $\mathbf{P}_{init}$ is then defined as the peak-normalized PM-spectrum multiplied by the data variance $\sigma(\mathbf{b})^2$ plus a 10% noise floor (Fig. A1c black dashed line). The initial prior is used to perform a first inversion of this segment, but – to avoid over-fitting to the PM spectrum – a seconds inversion with the same data is done, but now with a "data" prior, i.e., the smoothed result of the first inversion (Fig. A1c black line). The power of the 2nd (final) GFT model coefficients $\hat{\mathbf{p}}$ is then shown in green figure A1c).

Segments with a successful inversion in the previous segment do not make use of the PM-based prior; Instead, they use the data prior from the previous segment: For a segment $i$ with a successful inversion in the previous segment $i-1$, we use a smoothed power spectrum based on $\hat{\mathbf{p}}_{i-1}$ to derive $\mathbf{P}_i$ (Fig. A1e and g, black lines, sec. 2.4). Note that even though the initial PM-prior pushes the model to a single peak spectrogram (Fig. A1b and c dashed line), in cases where the data does not support this shape, as in this example, the PM-prior does not determine the result (Fig. A1c,e,g green lines). Any spectral shape, including double peaks, can appear. Cases where a PM prior improves the inversion are shown in suppl. Figure S12.

## Appendix B:  Wave-watch III Prior

The prior in eq. 12 uses an incident angle $\theta_0(k)$ with an uncertainty $\sigma_\theta(k)$, defining the prior

$$P_\theta(\theta, k) = \left( \frac{\theta_0(k) - \theta}{\sigma_\theta(k)} \right)^2. \tag{B1}$$

Both variables in the prior have to be taken from other data sources than IS2, and here they are derived from WW3 global hindcast wave-partitions (Tolman, 2009, using the Integrated Ocean Waves for Geophysical and other Applications (IOWAGA) hindcast)) and depend on wavenumber. The WW3 hindcast data is selected in a box around the most equatorward photons in ICESat-2 (suppl. Fig. S11 and. sec. 2.1). In cases where this box is within the sea-ice mask of WW3, it is moved equatorward along the IS2 track until at least $2/3$ of the box's grid cells are not covered by the WW3 sea-ice mask.

Within each box, the mean of the peak period, peak direction, and directional spread is calculated for each of the five WW3 wavenumber partitions (Fig. 7a black dots). These partitions are interpolated and smoothed to the $k_i$ wave numbers of interest to best guess the wavenumber-dependent peak direction and spread (Fig. 7 an orange line and shading). Note that it is hard to validate the WW3 partitions in the Southern Ocean due to a lack of contemporaneous in-situ observations. The lack of certainty in the wave-hindcast, in combination with (any) smoothing procedure, can lead to biases in the directional prior. Hence, we do not expect the direct alignment of the WW3 wave directions and those observed in ICESat-2 observations. The WW3 incident

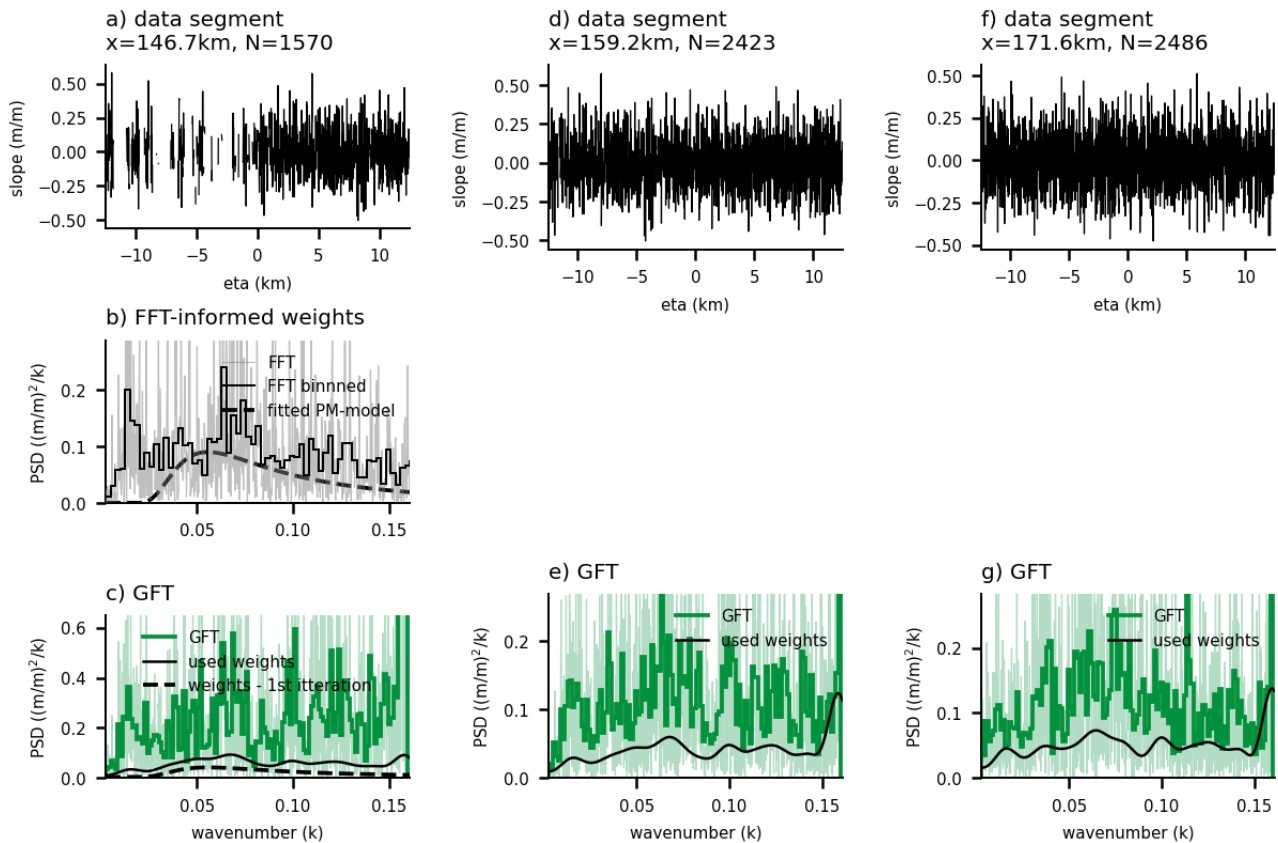

**Figure A1.** Examples of GFT inversion and their priors, for example, Track 1. (a, d, f) data used in the segment centered at $x = 146.7, 159.2$ and 171.6km. (b) DFT (gray), re-binned DFT (solid black) for data in (a), and PM-model fitted the DFT (dashed black line). (c, e, g) GFT (green line), re-binned GFT (thick green line), PM-prior for 1st inversion (dashed black line), and data-prior for second inversion (solid black line).

angle is here solely used to reduce ambiguities in the objective function (Fig. 6 b and d and Figure 7) and can give a *preferred* incident wave angle, rather than a certain estimate of the dominant wave angle (Fig. 7).

*Author contributions.* MH developed and programmed the proposed algorithm. MH and CH collaboratively wrote the paper.

*Competing interests.* We have no competing interests.

*Acknowledgements.* We thank Bruce Cornuelle and Baylor-Fox Kemper for discussing the depths and caveats of the proposed algorithms. C.H. and M.H. were supported by Schmidt Futures—a philanthropic initiative that seeks to improve societal outcomes through the development of emerging science and technologies. C.H. and M.H. acknowledge support from NASA 80NSSC20K0959 and NASA 80NSSC23K0935. C.H. acknowledges support from NSF OPP2146889.

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
