# Peer review of "A Method for Constructing Directional Surface Wave Spectra from"

_EGUsphere, 2022_

## Author Comment (AC1)

EGUSPHERE-2022-842
**Comments to RC1**
**Our replies are in blue**

My apologies for the long delay in returning this review. I understand that it is very unfair to wait for so long, but a few issues concurred to this delay, including the complexity of the manuscript itself. I would like to state from the beginning that I am not an expert in the specific techniques used by the authors, and that I have sought assistance with a colleague applied mathematician. I am however well informed on wave-in-ice measurements and modelling, and I feel this manuscript represents a major advance in the field.

*Thanks for your efforts, and we appreciate your kind words about the use of this work.*

It presents a two-part algorithm. The first is a linear harmonic fitting procedure applied to single beams. This part of the algorithm is almost fully explained by the authors and in the referenced literature. The main novelty here respect to literature is the use of GFT-derived surface slope spectrum. The second part of the algorithm is a non-linear inversion method to estimate the most-likely wave incident angles. It is the most articulated part of the paper, and not always well-structured. The approach is based on the cross-beam coherence, and uses marginal distributions derived from independent MCMC samples of monochromatic plane waves. The samples are limited to the most energetic waves. As commonly occurs for 2D spectra retrieval from satellites, the algorithm returns multiple potential incidence angles and therefore need of a good prior guess for directional spectra is needed. These combined procedures provide two-dimensional wave spectra derived from along-track data. The authors also discuss how the so-obtained two-dimensional wave spectra can be used to decompose wave and ice surface variability in Section 4, which becomes one of the main discussed components in the last section.

Despite the novelty, I found the manuscript difficult to read, and especially to fully grasp the implications of this method. I would thus recommend publication of the manuscript, pending some major revisions that would make it more accessible to a wider audience, or at least would clarify that it is directed at the technical audience that is interested in the retrieval of waves in ice from space. The introduction is excellent, but unfortunately not entirely connected to the discussion through the results.

*Thank you for these comments, and below we endeavour to respond constructively to all of them!*

I list below a few major points that the authors may consider in their revision:

1. My main criticism is that this is a methodological paper, which is however presented as a result paper. The results are functional to the methodology. From the title, I expected to be exposed to a reconstruction and interpretation of directional surface wave spectra from a variety of regions, while the manuscript is focused on 4 selected tracks from the Southern Hemisphere, each pair coming from two consecutive tracks in February and

May. By the way, the reader would find out that the focus is on Antarctic sea-ice waves only from the supplementary table or at L85, where Antarctic is mentioned for the first time. This is not at all to diminish the value of the manuscript, because a methodology must be demonstrated by example, and I regard the work done by the authors of very high value. I would suggest that the title reflect that the reader will find a description and application of the methodology, rather than expecting a thorough analysis of 2D spectra.

We have now changed the title to reflect that this is a methods-focused paper, as well as the abstract.

I would also recommend to add a description/discussion of the results presented, which is currently left to the readers. For instance, Fig. 5 and Fig. 8 show the power spectra from one example track from May 2019. This is the main result of this paper, but it is offered with very little interpretation or discussion in Sec. 4.

We now added more analysis in section 3.1 (L.290f).

> *"An example of the resulting beam- and wavenumber-average PDF is shown in figure~\ref{fig:mcmc2}b for Track 3 at $X_i=87$. Here, the individual PDFs are weighted by the mean power of the respective wavenumber and the number of data points per segment pair. The most likely incident angle is at $-37^\circ$, while two other angles $-63^\circ$ and $0^\circ$ also show high likelihood. Marginal PDFs with multiple maxima are a typical result for this method and appear in many other tested sections and tracks (not shown). They come from different maxima in the joint PDFs of different wavenumbers. If one trusts the angle estimate of a single wavenumber, this result can be interpreted as a wave field with waves from multiple directions. The alternative -- likely better -- perspective is that the marginal PDF of a single wavenumber is not a robust estimate of the incident angle, and hence the PDF in figure~\ref{fig:mcmc2}b helps estimate the uncertainty of the method."*

And in section 3.2 (351f):

> *"Past the primary wave event, a second signal further into the ice with energy on scales shorter than 200 meters extend from $X>75$ km about 100 km or more (Fig.~\ref{fig:final_corrected}a,b). This signal is at shorter wavelengths than the identified cut-off frequencies for the event at the track beginning. Without further information about the ice conditions, we suggest that this short-wavelength energy deeper in the sea ice is due to sea ice variability itself rather than due to waves"*

Only a brief comment about the attenuating swell signal and a second energy maximum at shorter wavelength (L200-201), and similarly, a migration of the peak due to the attenuation of shorter waves at L316-320. The findings seem to me to be coherent with the local observations by Alberello et al. (2022), who differentiated between wind sea and swell. Almost no discussion is offered on these phenomena, while the matter of the

discussion is mostly about the methodology and its limits (again, I find it very well done and interesting and not at all a fault if properly framed).

We specifically reference Alberello et al. (2022) in line 349 and in the discussion (L476). We decided not to mention the difference between wind sea and swell since we cannot differentiate the wind sea from swell without additional wind or wave data on hand, which we note on L421: *"Without adding observations other than ICESat-2, we are not able to directly differentiate between wind sea and well."*

2. Alberello et al. (2002) found that the wave field along a transect in the Antarctic MIZ was composed of wind and swell from different directions. With this procedure, a unique most likely incident angle is estimated. This implies that both swell and wind waves propagate in the same direction, which may not be true. The authors should consider this in their discussion.

We thank the reviewer thoughtful comment about multidirectionality. In fact, the MCMC method has a frequency dependent prior to using the WaveWatch III hindcasts partitions (Appendix B).

However, the signal-to-noise ratio of the MCMC method may not allow for more than one incident angle to be used for the angle correction of the power spectra. Hence, a correction of the observed wavenumber that accounts for the heterogeneity of the wave incident angles in sea ice (Alberello 2022) may not be possible without better priors. The current lack of these priors lets us assume one dominant wave direction for the entire spectrum, a justified assumption since the method aims to detect the attenuation of the dominant wave energy. We clarified these aspects in section 5.2, line 475f:

> *"The angle inversion method generally can sample multiple incident angles, i.e multiple minima in the objective function, and may be able to detect multiple wave systems from different directions \citep{Alberell2022Threedimensional}. However, after testing the effect of wave groups and uncorrelated noise on the sampling method, we came to the conclusion that the signal-to-noise ratio is too low for a frequency-dependent angle correction, even after adding a prior. Hence, for the purpose of angle correction, we here assume that all wave energy comes from the same angle, i.e. the angle of the most energetic waves. Even though the true wave field may be a superposition of multiple wave systems with varying directions, the single incident angle is justifiable here, because we are focused on the attenuation and propagation of the dominant wave energy."*

We also add in the abstract (l. 11) an introduction that this method is useful predominantly for resolving the most significant wave energy for the purposes of recording wave attenuation (l. 61).

3. As of the ice structure mentioned in the title, this is mostly a potential application that is not exploited in the present manuscript. Such a component would require a more

thorough validation that has not been included. It is not clear if the considerations on the structure of sea ice should come from Sec. 4. This leads to the interesting discussion in Fig. 10 and 11, but I do not see how this would inform about the ice type.

Thanks for that observation. We have now removed the reference to sea ice structure in the title, in section 4, and throughout the text. We also edited the short discussion about using the residual heights in section 4 (L388f):

> "… . If the contribution of non-wave signals to the across beam average is minor, this decomposition removes waves as the dominant source of variance on scales larger than 20 meters. This allows for additional analysis of the residual signal, and more consistent surface height signals in wave-affected and low-sea ice regions. A better filter design can further improve this separation between waves and sea ice."

4. Also related to Sec. 4, in Figures 9 and 10, k_c ranges from 0.05 to 0.1. This is the range of wind waves. Therefore, the residual z_{free} (defined after Eq 12) should estimate both the sea ice freeboard height and the surface wind waves. It seems that the low pass filter can only remove the contribution due to swells. The authors should elaborate on this in their applications of the method.

Indeed, for cases with a wind sea and complex sea ice structure, this method will not be able to clearly separate both. Only the addition of other data may help to identify the signals. We discuss this now in L 417f:

> "The chosen examples show a clear wave signal that can be separated from high-frequency noise by using a simple low-pass filter with a cut-off frequency $k'_c$ (sec.~\ref{sec:variance}). This cut-off frequency assumes a scale separation between waves and sea ice variance, that generally may not exist. The identified cut-off frequency lies in the plausible range of wind waves ($k=0.05$ to $0.1$). In cases with wind waves and complex sea ice structures \citep{Alberell2022Threedimensional}, a separation between the wave and sea ice signal is hampered. Without adding observations other than ICESat-2, we are not able to directly differentiate between wind sea and well."

5. I like Figure 1, which is well done and illustrative. However, and partly related to the first point, it would be useful to have an example of a track location, together with the configuration of sea ice (for instance sea-ice concentration). This could be done for either the February or May cases in Supplementary Table 1. The May tracks are from the same day, so it may be easier to show them on the same map with the daily SIC. It is also the track used to show the spectra in Fig 6 and 8.

*We add a panel in Figure 1 that shows the SIC and the IS2 tracks for the May case.*

[Figure]

a) ICESat-2 ATL03 data (track: 05160312) and CDR Sea Ice Concentration on 2019-05-02

b) Wave Observations by ICESat-2 in the Marginal Ice Zone

c) Schematic of the observed phase lag and incident angle by an ICESat-2 beam pair

**Figure 1.** Illustration of ICESat-2 (IS2) beams intersecting the Marginal Ice Zone (MIZ) under the presence of waves. a) ICESat-2 data and the CDR Sea Ice Concentration for track 05160312 on May 2nd 2019. The three beam pairs are shown as red/orange (gt1l/gt1r), dark/light green (gt2l/gt2r), or dark/light blue (gt3l/gt3r) lines. The black dots show the segment positions 12.5 km apart (define in section 2.3). b) Schematic of the IS2 beams that observe sea ice and surface waves in the MIZ. The vertical black line is the nominal IS2 reference ground track. The incident waves come from the top-right along the black arrow with wave crests (solid) and valleys (dashed). c) Details view of an ideal monochromatic wave observation by IS2 in sea ice. The IS2 beam pair (light/dark green lines) observes an incident wave (along the black arrow) of wavelength $\lambda$ with an angle $\theta$ as $\lambda'$. With the beam-pair distance $d$, one can calculate $\theta$ from the phase lag of the incident wave crests (sec. 3.1). The along-/across-track coordinate system is referenced to the nominal track, while each data point is the weighted mean of a 20-meter stencil (sec. 2.1).                                    `f:fig02_geometry`

6.  This method will require an independent set of assessments before being considered viable. There are many subjective (though reasonable) choices that the authors have made both during the pre-processing phase and in the method implementation. This may have implications that are hard to assess without a benchmark. I understand that these datasets of combined wave measurements in sea ice and in the adjacent open ocean do not exist, and the few that are available are prior to ICEsat2, or are not done in the vicinity of available tracks. However, the method is being sold as fully functional, which I agree from a technical viewpoint, although it is still doubtful whether it is reliable. This should be made clearer in the discussion.

Even though we decompose the complete variance field, it will be interesting to see how this method compares to in situ observations. We add text in the abstract (L14f) conclusion (L 507f) to emphasize this is an initial deployment of this method that indeed needs further validation:

> *"Even though this method outlines a better, more transparent wave-field inversion than a DFT, it remains to be seen how the interaction of those limitations can be used to provide a highly-resolved global wave-in-ice product.*

*Comparisons with other data sources, either from in-situ or remote sensing observations, are needed to understand these limitations better and validate this method."*

7. This manuscript contains quite a few English inconsistencies, as though it has been submitted without thorough proofreading – possibly from someone else besides the authors. It is clear that the authors have a good mastering of their methods, but this makes the explanation often convoluted and difficult to follow. I have included a few specific recommendations below when I felt I understood the intent, but I would advise the authors to take a slightly more distant approach and focus on the information transfer rather than on the flow of concepts.

*We have read through the manuscript to generally improve the flow and language in the text. Specific suggestions would be helpful on a next revision if the reviewer still feels there are challenges with the writing.*

**Specific comments**

- Sec. 2. Figure S1 is not mentioned in this section but later in the manuscript.

*Thanks for pointing this out. We re-assigned the suppl. Figure numbering according to their appearance.*

- L 82-83 Some more clarity on the pre-processing method would help: is there a rationale for the choice of 0.02 photons per 100 km? Is this done prior to stencils determination explained at L94-95

*This is a typo, and we set a threshold of 0.02 photons per meter estimated over a 100km segment. This is a very low threshold on purpose to just exclude areas that have nearly no data. This threshold is also independent of the minimal photons per 20-meter bin. We clarify that in the text (L 84f):*

*"The most equatorward position is evaluated from the ATL07 \citep{Kwok2021ATLAS} product and set to the beginning of the 1st 100 km of along-track data where there is an average of at least 0.02 photons per meter (defined as $X=0$ throughout the paper)."*

- L96 The photon height in each 20 m stencil is a series of heights. Maybe the authors refer to the photon height mean

*We now use mean photon height in L101, 107, and throughout the text.*

- L99 This must be S3. And what are the data shown on the x-axis in that figure? Is \sigma_x the standard deviation?

*After re-ordering the Supplementary figures, this is now indeed S2.*

- Figure 2. There is a typo in ATL03 and ATL07. What is the offset of the grey line?

*Thanks for catching the typos. The gray and red segments show the ATL07 classification. Red is a sea ice lead, and therefore ssh classification, and gray is a sea ice classification. We clarified this in the caption of figure 2.*

- L149 misleading

    - fixed!

- In Eq 8 and Eq 9 data are assumed Gaussian. Nevertheless, also the authors recognize that data are not Gaussian in Sec 4.

We thank The reviewer for the comment. Yes, that is a limitation in using a linear model. We decided to use surface slopes rather than total heights to mitigate the effects of sea ice heights or secondary wave effects. We made a short comment in the discussion section 5.1 (L440):

    *"As with any linear inverse method, the GFT assumes Gaussian statistics, which is obeyed by linear waves but potentially violated by sea ice surface variability. To minimize the effect of sea ice heights on the wave inversion, we use mean surface slopes rather than heights (sec.~\ref{sec:prehandling}), which results in an approximate Gaussian residual (Fig.~\ref{fig:GFT_result}e)."*

- In subsection 2.3.1 model's degrees of freedom are treated. How many times the problem remain overdetermined, i.e what is the mean value for N_i?

This depends on the sea ice's sparseness in the MIZ. We only try inversion where more than 10% of the possible 2500 data points exist. We show the new supply. Figure S3 (see below). Most of the segments in the 4 chosen tracks are overdetermined. We made a short comment in L185f:

    *"Most of the segments of the four example tracks in this study have close to 2500 data points and are over-determined (suppl. Fig.~S3). Only track 1 and 2 (Fig.~\ref{fig:GFT_result}) are under-determined close to the edge of the ice cover (supply. Fig.~S3 a,b)."*

[Figure]

- Figure 3 is not discussed in the paper, specifically the peak observed in panels b and d for k>0.06. Are these the same beams shown in Figure 5? Also, the spectral energy for each wavenumber is positive in Fig. 3 but it is negative here.

- Figure 3 and 5 show data from different tracks to show how the algorithm deals with data gaps. The data in Figure 3 is presented as the spectral amplitude of the (slope^2/k), while the data in Figure 5 is the power of the slope 10*log(slope^2/k), which leads to negative values. We added a description of the results at the end of section 2.3 (L171f ):

> *"The harmonic inversion of this segment of Track 2 shows how wave spectra can be calculated from strong (gt2r) and weak (gt2l) beams, even if the data has gaps (Fig.~\ref{fig:GFT_result}a). Both beams' spectra are similar in most parts and show a maximum at $k=0.03$ (Fig.~\ref{fig:GFT_result}b,c). However, the strong beam (gt2r) does show a second local maximum at about $k=0.06 $. Note that the DFT of the same track, and with tapered data, results in a different PDF than the harmonic inversion because of the data gaps."*

- L200 is X the distance from the ice edge or from the first point on the track? It is ambiguous because most of the axes labels indicate "distance from the edge". And if yes, how is the edge estimated?

X=0 is a point on the equatorward side of the track where the photon density exceeds 0.02 photons per meter. We introduce this in L84f:

*"Along-track photon positions are first re-referenced to the most equatorward position on the nominal ATLAS ground track (Fig.~\ref{f:fig02_geometry}b, black line). The most equatorward position is evaluated from the ATL07 \citep{Kwok2021ATLAS} product and set to the beginning of the 1st 100 km of along-track data where there is an average of at least 0.02 photons per meter (defined as $X=0$ throughout the paper). This threshold and re-referencing are used to exclude large areas of nearly no data in the transition zone between the open ocean and MIZ."*

And also explain it in L215f, and in the caption of Figure 5.

- L201 should be a typo: wavelength is indicated as k'.

This is the observed wavelength k'. We clarified that when introducing the notation in L140f:

*"We use prime notation here and throughout the paper to indicate observed wave variables along the direction of each beam and unprimed notation for variables in the direction of the traveling wave… "*

- line 213 "k = k'cos^{−1}(\theta)", and hereafter: cos^{-1} is also used to indicate the arccosine. It is better to use a different notation. Moreover, the equation needs a reference (eg Yu et al., 2021)

*We changed the notation to \cos{(\theta)}^{-1} throughout the text.*

- at the beginning of page 12 "The coherence between beam pairs can only usefully be calculated for not too oblique angles ( about ±XX°, Suppl. Fig. S3, and Yu et al., 2021) and high enough photon densities in both beams." The figure should be S4 which is also not so readable. It is also unclear what \pm XX° means, please add some specific examples.

We clarified the statement and limited the analysis to +- 75 degrees. More oblique angles can hardly be inferred, given the distance between the tracks and the dominant wavelengths (L240ff):

*"The phase lag between beam pairs can only usefully be calculated for not too oblique angles \citep[Suppl. Fig.~S5, and ][]{Yu2021Assessment} and high enough photon densities in both beams. The angle limits in which the phase lag can be resolved depend on the chosen wavelength and the distance between the beams. Since both, wavelength and distance, change for each segment, we here limited the analysis to angles of $\pm 75^\circ$."*

- Eq. 11 should specify the cost function even without the prior (I guess set P=0)

*In the interest of brevity, we presented it with the regularization term here and hope this is acceptable to the reviewer!*

- L268. How different is this track example from the other cases? Since there is no discussion on the results, it is difficult to imagine how different are the results in February and May.

Thanks for the suggestion. We edited the paragraph to describe the result better and put in context (L290f):

> *"An example of the resulting beam- and wavenumber-average PDF is shown in figure~\ref{fig:mcmc2}b for Track 3 at $X_i=87$. Here, the individual PDFs are weighted by the mean power of the respective wavenumber and the number of data points per segment pair. The most likely incident angle is at $-37^\circ$, while two other angles $-63^\circ$ and $0^\circ$ also show high likelihood. Marginal PDFs with multiple maxima are a typical result for this method and appear in many other tested sections and tracks (not shown)..."*

- L281 and following. The authors speak about statistical robustness. The meaning should be explained

We changed the wording to "a good estimate of the mean incident angle and its standard deviation" in L304.

-L295-296 This is related to my main point above. The authors state that the effect of the prior is visible in the figure. But there is little comment and no discussion.

See our reply to your main point 1), the added discussion at the end of section 3 (L341 to 354), and especially the added discussion in section 5.2, L482 to L488:

> *"The low signal-to-noise of the angle inversion requires regularization (sec.~\ref{sec:angle}). Since directional wave observations co-located with IS2 tracks are sparse and not readily available, we relied on Wave Watch III (WW3) hindcast models as a prior \citep[IOWAGA][]{Tolman2009User}. The wave hindcasts may perform sufficiently well in the Northern Hemisphere but are known to have limitations in the Southern Ocean MIZ, potentially due to wind biases \citep{Belmonte2019Characterizing,Hell2020Estimating,Hell2021TimeVarying}. The lack of certainty in WW3's peak direction and frequency is expressed in the value of the hyperparameter $\beta_\theta$ (eq.~\ref{eq:angle_cost}). A value of $\beta_\theta=2$ leads to the desired behavior of breaking the symmetry (compare shading in figure~\ref{fig:mcmc} b and d) but not imposing the optimization result through the prior (Fig.~\ref{fig:mcmc2}d blue and orange lines)"*

- line 306 "This method is limited to angles of about ±85. Oblique incidence angles cannot be captured by this method. In addition, the model has a 180 ambiguity such that samples in the +95 to −95 arc, that are waves coming from higher latitudes rather than from the equator, would be equally possible", please clarify.

We rewrote the paragraph *(L332ff):*

*"This method is limited to angles of about $\pm 75^\circ$ deviation from the nominal track direction. More oblique, i.e. steeper incidence angle can not be captured by this method because a steeper angle requires more coherence between wave crests. The coherence of a single wave crest is, however, limited by the curvature of the wave spectrum and not well known (suppl. Fig.~S7). In addition, the model has a $180^\circ$ ambiguity such that waves coming from the equator side of the track (as assumed), or waves coming from the pole side (less likely) can result in the same phase lag and hence in the same incident angle, even though they come from the opposite direction."*

- Line 318: "In the case of Track 3 (granule 05160312) for example, we see a migration of the peak wavelength from about 275 meters to about 300 meters within 12.5 km (Fig. 8 d,e) as shorter waves tend to attenuate faster". Colorbar in Figure 8 is missing and the sentence is not clear enough to estimate wave attenuation.

We here outline that this method potentially can be used to measure wave attenuation and would be able to observe a migration of the peak frequency, but not estimate wave attenuation yet. We added a colorbar to Figuer 8 and rephrased the statement (L346) :

*"In the case of Track 3 (granule `05160312`), for example, we see a wave event coming from about 45◦ to the right of the ground track that mostly attenuates in the first 75 km from the sea ice edge, while the overall attenuation rate is similar between the six beams (Fig. 5a-f). One could identify a migration of the peak wavelength from about 275 meters to about 300 meters within 12.5 km (Fig. 8 d,e, similar to Alberello et al., 2022); we leave this analysis of the attenuation to future work."*

- L373 It is not clear what the authors mean with improved understanding of photon variance. Also, the authors could comment on why the variance is so large in the interior of the MIZ, especially in track 2 and 4, but not in Track 3.

We edited the sentence in L411:

*"Our quantification of wave energy allows for an improved understanding of the observed surface elevations in sea-ice-covered areas. We showed that ocean surface waves have an important contribution to the variance in the MIZ."*

*We also decided not to show the anomalous variance further into the ice in figure 11, as we think this has to do with other sea ice structure like rafts and cravas that are not in the focus of this analysis.*

- points at L395. This is a methodological paper, and I find quite ambitious that all of this can be achieved by a non-validated methodology

We rephrased these statements and moved them to the conclusion (L526f):

*"This will be used to statistically constrain parametrizations of wave-attenuation in sea ice (Fig.~\ref{fig:GFT_2}) and can potentially leverage the wave-removed residual signal to improve ice classification (Fig.~\ref{fig:decomp} d)"*

- In a few figures and captions the authors often refer to something undeclared, or declared only in supplementary materials (eg: FFT, granule, gt1l,...)

Thanks for mentioning this. We made sure that we introduced the term granule in line 67 and other acronyms throughout the text.

*"... Track 1 to 4 (details in suppl. table. Their granule, i.e. their identification number, is also given in each figure) …"*

**References**

Alberello, A., Bennetts, L.G., Onorato, M., Vichi, M., MacHutchon, K., Eayrs, C., Ntamba, B.N., Benetazzo, A., Bergamasco, F., Nelli, F., Pattani, R., Clarke, H., Tersigni, I., Toffoli, A., 2022. Three-dimensional imaging of waves and floes in the marginal ice zone during a cyclone. Nat Commun 13, 1–11. https://doi.org/10.1038/s41467-022-32036-2

---

## Author Comment (AC2)

EGUSPHERE-2022-842

**Comments to RC2**

*Our replies are in blue*

The manuscript presents a method to estimate the directional wave spectrum from satellite measurements taken in ice. As I understood, the method has two parts. First a GFT is used to calculate the slope spectrum, which will be a function of the along track wavenumber. Second, the coherence between different tracks are used to estimate the incident angle of the waves in relation to the track. This angle gives directional information, but since the relative angle is also used to transform along track wavenumbers to actual physical wavenumbers, even an omnidirectional spectral estimate is dependent on both parts of the algorithm. Finally, the authors remove the wave dependent elevations from the signal, thus getting an estimate of the ice properties only.

I find the topic of the manuscript to be novel and interesting. The authors have generally made a good effort in presenting the method and the choices they made. I still found some parts hard to follow, especially since all of the notations seemed to be not defined. While the method seems reasonable, it is very hard to actually judge the accuracy and limitations of the method and the following results without any other data to compare it with. I still think this methods paper is worth publication, but I do feel the authors should make it clear that the method is still not properly validated.

Thank you for these comments. As noted in our response to reviewer 1, we have endeavored to re-frame the paper as being methods-focused. We now additionally add text in the abstract and conclusion about the need for validation, in L507f:

> *"Even though this method outlines a better, more transparent wave-field inversion than a DFT, it remains to be seen how the interaction of those limitations can be used to provide a highly-resolved global wave-in-ice product. Comparisons with other data sources, either from in-situ or remote sensing observations, are needed to understand these limitations better and validate this method."*

and L513ff:

> *"However, even on these scales (80 to 350 meters), a separation between wave and sea-ice signal may only be possible when the sea ice variance is weak on those scales and the data is not too gappy, as in the chosen example tracks (Fig.~\ref{fig:GFT_result}, Fig.~\ref{fig:decomp}). High levels of sea ice variance or frequent data gaps will lead to systematic biases and aliasing effects in the wave spectral estimates."*

I can recommend that the manuscript is accepted for publication after major revisions. Please find my comments below:

**1 Not all of the notation is defined. Eg. on page 7 hat-b on line 146 is not defined, and "tr" is not defined on line 156.**

*Thanks for pointing this out. We went over all notations and made sure variables and operators were defined (section 2.3)*

**2 The shortest wave being resolved is about 60 m, and this should be well resolved with a 10 m resolution. However, if we have missing data the actual resolution could be lower, and this would induce aliasing. Based on the figures presented the gaps seems to be few and long instead of numerous and short, so this is probably not an issue though.**

Yes, that can be a problem, but as the reviewer mentioned, for "small" gaps, that should be okay. Note that fewer data points will generally lead to larger error estimates. We assume that the wave field is smooth in space and time, which implies that it may vary over larger scales than a 25 km segment and that the energy in one way number should be similar to its neighbors. That means that the estimated spectra should be similar in neighboring beams and neighboring 25km segments(Fig. 5). If they are not, that is a sign that the data structure is too *complex (too gappy) for this method. See the added sentence in L 223f:*

> *"Instances with a low photon density and more frequent data gaps may fail to invert for the wave signal, resulting in a spectrogram that may not follow expected spectral shapes. These can be identified through their substantially larger error (suppl. Fig. S4 g to i).",*

*and L450ff:*

> *"A major advantage of the GFT is that it can be extended to inversions of the wave field for each IS2 track by coupling neighboring or overlapping segments, similar to Kalman inversion methods. We illustrate this by simply iterative updating the data segments and models priors (sec.~\ref{sec:suggsesive}, Fig.~\ref{fig:GFT_alter}). …"*

**3 Page 8 lines 181-182 mention that the Pierson-Moskowitz spectrum is taken as an initial guess at the outer edge. While this is reasonable, I worry how much the initial guess of a single peaked spectrum will affect the way the method resolves a double peaked spectrum?**

*That is a great question, and we were concerned about this as well. We used a single peak prior because we focus on the inversion of the dominant (i.e., most energetic) wave system, as those may have the most impact on the sea ice. To avoid overfitting to the PM, we perform two inversions on segments with no data available at previous segments (see Figure 4). As we now clarified in the text (L 205f) and Appendix A2, for initial segments, the 1st inversion is based on PM, and the 2nd inversion is based on the smooth data of the 1st inversion. Tests show that the initial PM-prior has only weak to no influences on the solution of the initial or following segments for cases with no single peak spectrum (updated figure A1) but helps to retrieve a single speak spectrum for cases with gappy data (Suppl. Figure S12). We show here the new Figure A1 for convenience:*

[Figure]

**Figure A1.** Examples of GFT inversion and their priors, for example, Track 1. (a, d, f) data used in the segment centered at $x = 146.7, 159.2$ and 171.6km. (b) DFT (gray), re-binned DFT (solid black) for data in (a), and PM-model fitted the DFT (dashed black line). (c, e, g) GFT (green line), re-binned GFT (thick green line), PM-prior for 1st inversion (dashed black line), and data-prior for second inversion (solid black line).

**4 Figure 3 mentions that the spectrum is smoothed by a rolling mean. **First, the rolling mean is perhaps not a very good filter and could possible introduce artefacts.** Second, noise in spectral estimates are usually accounted for by either splitting the time series into blocks (the Welch method) or averaging neighbouring wavenumber bins (theoretically these are equivalent). So I think the right thing to do here is to **average wavenumber bins.** This will reduce noise, but also the resolution. That way, however, the resolution will be an accurate account of the true resolution of the spectrum. Using a rolling mean means that the resolution appear higher, but the bind are no longer independent.**

*Thanks for the comment. We now present the estimated spectra (Figure 3b,c,b) and the weighted cross-beam mean spectra (Fig. 5h, Fig. 8a,b) as a binned average. It does not alter the overall result.*

*We here how the new Figure 3b,c as example*

[Figure]

b) Beam gt2l Spectral Estimate

c) Beam gt2r Spectral Estimate

wavenumber k ($2\pi$ m$^{-1}$)

**5 page 10 line 212. I think cos-1 would usually mean the inverse function here? Maybe better to just use k'/cos to avoid all confusion.**

*Thanks for the suggestion. We changed the notation accordingly.*

**6 The manuscript talks about using coherence to estimate the angle (e.g. page 12 line 220). I'm not sure how this is done using coherence. I would think that the phase lag between the signals would be used to determine the direction, and the coherence would only tell us if the phase lag measured from the corss-spectral estimate is "real" or not. This method reminds me a lot of the Wavelet Directional Method (see Donelan et al., 1996), although on a wildly different scale. This method has been used by the wave community, and I think it would be a good idea to point out the similarities and differences to this more established (although not extremely widely used) method.**

Thanks for the suggestion. Indeed the use of the term spatial coherence is misleading in the context of spectral analysis. In section 3.1 and throughout the text, we now use the term '*phase lag*' when discussing the angle inversion method.

We appreciate your suggestion of the WDM. There are indeed some similarities between both approaches. In fact, the approach the measure the phase lag per wavenumber is the same in both methods. Both operate with the same amount of information per observation, but the main difference is that WDM requires three observational time- or spatial series, while ours has to operate with only two. A 3rd laser beam close by would allow for applying the WDM. However,

the ICESat-2 laser beams are too far apart (about 3 km) such that we have to introduce a prior for breaking the symmetry. We briefly discuss this in section 5.2 line 489ff:

> *"The proposed MCMC method shares aspects with the Wavelet Directional Method (WDM, Donelan et al. 1996, 2015), which decomposes the signals of at least three stationary wave observations into wavelets for each frequency. Similar to our method, WDM uses the phase lag of the wavelets between the three stations to identify a wave incident angle per frequency. WDM could be applied to transects of the wave surface as present in our analysis. However, ICESat-2 only provides two neighboring laser beams, and other beam pairs are too distant (about 3 km) for coherent phase analysis. In addition, the signal-to-noise may be substantially lower in the ICESat-2 observations, as wave crests are potentially distorted by sea ice structure. Therefore, we introduced a wave-angle prior (eq.\ref{eq:angle_cost}) to break the ambiguity in the observed phase lag (Fig.~\ref{fig:mcmc}b,d, shading)."*

**7 Connected to the above, I would think that several angles would result in the same phase lag (e.g. a small angle gives half a wavelingth lag and a large angle 1.5 wavelength lag). Is this what the authors refer to on page 14, line 293 with "multiple equally likely incident angles", or are they talking about some other source of uncertainty? I think it makes sense to use the longest waves to determine the angle, since they are least sensitive to this kind of "folding".**

This method has to deal with multiple symmetries that we try to mitigate. In this case, we describe the problem that there can be a positive and negative leg, i.e., the waves can come from left or right. In addition, there is also the 2 pi ambiguity that leads to multiple minima in each quadrant. We clarify this in the updated manuscript L.317f:

> *"This joint distribution may have multiple equally likely maxima, i.e. multiple likely incident angles due to the periodicity of the wave ($2\,\pi$ ambiguity). As illustrated in figure \ref{fig:mcmc}d (shading) this can lead to a) maxima for positive and negative incident angles and b) multiple maxima on both sides."*

Longest wavelength would work better, but they are not necessarily the most energetic. That is why we speculate that this method may work best away from the ice edge when short waves have attenuated. We describe these challenges in L 466ff:

> *"The quality of MH inversion method depends on the wavelength, wave amplitude, and curvature of the wave spectrum. The longer the wave the better the phase lag can be observed, but those are not the most energetic. In turn, the most energetic waves have typically shorter wavelengths that are of 80-250 times the segment length (25km), which can lead to multiple minima in the optimization due to a $2 \pi$ ambiguity. Finally, the curvature of the wave spectrum characterized the length of wave groups, which in 2D, erode the ability to observe the average phase lag between the two beams (2nd bullet point in sec.~\ref{sec:angle}, suppl. Fig.~S7)."*

**8 Page 16 line 319 mentions a "migration of the peak" from 275 to 300 m. However, from Fig. 8 this seems to be the observed (along track) peak, and not the actual peak wavelengt. In the corrected one there doesn't seem to be much migration. But Fig. 9 shows spectra with peaks closer to the 0.02 rad/m, i.e. 300 m range. So am I reading this wrong somehow?**

We understand the confusion. Figure 8 shows the corrected slope spectra, and Figure 9 shows the uncorrected height spectrum. Both figures have different purposes. Figure 8 illustrates the angle correction and is based on Figure 5, while Figure 9 shows how the frequency cut-off k'_c is defined and used to reproduce surface heights. We clarified this in the caption of figure 9 and changed k_c to k'_c throughout the text, and when introducing figure 9 (L362f):

> *"In Figure \ref{fig:cut_off} we show the identified low-pass filters and the displacement spectrum (m^2 k^{-1}) rather than the slope spectrum ((m/m)^2 k^{-1}), as in Fig.~\ref{fig:final_corrected}) to better separate the high-frequency noise from the lower-frequency waves."*

We now also show the corrected wavelength in Figure 8 c) to d), and after reading the data, we weakened the statement of peak migration L. 348f:

> *"One could identify a migration of the peak wavelength from about 275 meters to about 300 meters within 12.5 km (Fig.8 d,e, similar to Alberello et al., 2022) we leave this analysis of the attenuation to future work."*

**9 **I am slightly skeptical to the last part of the method that removes the wave signal and estimates the ice surface roughness.** I might be missing something, **but isn't the scales of the surface roughness that can be resolved determined by kc?** I other words, when removing the wave signal for k<kc, we are also removing any possible NON-wave related signal in that wavenumber range. The final roughness (and the scales that will be represented) will therefore depend on which wavenumbers happen to have wave energy. As a corollary, the **wave spectrum also includes the variations of the changes in ice roughness over the same scales, right? Won't that taint the spectral estimate?**

Yes, these are limitations of the method presented. The use of a simple cutoff frequency assumes a separation of scales between longer wave scales and shorter "roughness" scales. There can clearly be cases where a scale separation may not be correct. This and the need for in situ validation are stated in the conclusion section, for example in L507ff:

> *"…, it remains to be seen how the interaction of those limitations can be used to provide a highly-resolved global wave-in-ice product. Comparisons with other data sources, either from in-situ or remote sensing observations, are needed to understand these limitations better and validate this method.*
>
> *Waves and sea ice have scales ranging multiple orders of magnitude such that it is challenging to separate both in the IS2 observations. The choice of the parameters in this analysis (10-meter bins, 25km segment length, and the slope-based cut-off frequency $k'_c$) focus on identifying swell wave events routinely created by synoptic*

*storms \citep{Hell2021Swell}. However, even on these scales (80 to 350 meters), a separation between wave and sea-ice signal may only be possible when the sea ice variance is weak on those scales and the data is not too gappy, as in the chosen example tracks (Fig.3, Fig.10)."*

In future work, one could think of including a model of sea ice roughness or simply height in the inversion method to resolve the sea ice structure explicitly.

**10 The manuscript has a discussion, but no conclusions. This leaves the reader hanging a little bit. I would strongly urge the authors to **conclude their findings**, and maybe also make it clear that this is still an **non-validated method**. Even if the theoretical approach seems sound to be, and a lot of the choices made by the authors seems reasonable, **it is very hard to say how much e.g. the values of the first prior spectrum or the value of the beta parameter actually affect the results.** To be clear, I'm not saying that the authors need to get data and validate, since such data is probably not readily available. I'm saying it should be made very clear to the reader that this has not been done.**

We added a conclusion section that emphasizes the need for validation and the choice of parameters to focus on swell waves (L507ff, same citation as above).

We share the reviewer's concern about the impact of the impact of priors and beta parameters. Those are always subjective to some extent, but we tried to make plausible choices. As discussed in the reply to #3, the GFT priors are chosen rather conservatively (see Appendix A2, suppl. Figure 11). Similarly, the beta-parameter of the angle inversion is chosen such that it just breaks the symmetry of the otherwise symmetric likelihood function, as shown in figure 6 b, d shading. The actual value of the prior angle does not matter much, it just gives a preference for one side, and the actual angle value is determined by the gradients in the data, not in the regularization. We clarify this in the main text L328ff:

> *Since validation of the WW3 prior is limited, we set $\beta_\theta=2$. Its effect on the objective function can be seen by comparing the shading in figure 6b and d. The choice of $\beta_\theta=2$ leads to the desired result in breaking the directional ambiguity while not fully determining the incident angle distribution (Fig.7a). We tested other values of $\beta_\theta$ but found empirically that higher values tend to overfit to the prior, and lower values do not break the ambiguity well.*

and L486f:

> *"The lack of certainty in WW3's peak direction and frequency is expressed in the value of the hyperparameter $\beta_\theta$ (eq.~\ref{eq:angle_cost}). A value of $\beta_\theta=2$ leads to the desired behavior of breaking the symmetry (compare shading in figure 6 b and d) but not imposing the optimization result through the prior (Fig. 7d blue and orange lines)."*

References: Donelan et al. (1996). "Nonstationary Analysis of the Directional Properties of Propagating Waves", JPO,
https://doi.org/10.1175/1520-0485(1996)026<1901:NAOTDP>2.0.CO;2

---

## Author Response (AR2)

**Response to reviewers II**

We thanks the reviewers again to reviewing the manuscript.

- We corrected the typos and reconciled the conventions for ICESat-2 (IS2) and WAVEWATCH III (WW3).
- L112: This was indeed at typo. we rewrote $k^2 \tilde{S}_h(k) = \tilde{S}_c(k)$

I also moved to a new position so it be good to add my current address to the manuscript:

National Center for Atmospheric Research, Boulder, CO, USA
1850 Table Mesa Dr, Boulder, CO 80305
mhell@ucar.edu